# Syk activation during FcγR-mediated phagocytosis involves Syk palmitoylation and desulfenylation

Maxime Jansen[1], Jean-Marc Strub[2,3], Laurent Chaloin[1], Peter Coopman[4], Bruno Beaumelle[1]

The Syk tyrosine kinase acts downstream of several immune receptors such as the FcγR. Syk owns two SH2 domains that interact with biphosphorylated ITAMs of the FcγR upon phagocytosis. This results in the activation of Syk by autophosphorylation, triggering phosphorylation of several downstream targets, F-actin polymerization, and phagocytosis of the IgG-opsonized target. We found that Syk is S-acylated upon phagocytosis by macrophages. Palmitoylation is performed on a single Syk-Cys by the DHHC5 enzyme that specifically associates with Syk upon phagocytosis. Syk palmitoylation is important for Syk localization to the phagocytic cup, phosphorylation, and phagocytosis. We observed that another Syk-Cys residue, within a redox motif, is modified by sulfenylation. Nevertheless, Syk desulfenylation seems to occur during phagocytosis, when $H_2O_2$ production at the cup decreases, after 3.5 min of phagocytosis. Molecular dynamics studies indicated that desulfenylation increased the exposure of a loop within the Syk interdomain B. This could facilitate phosphorylation of key Syk-Tyr residues by upstream kinases. We thus propose an updated model for Syk activation during FcγR-mediated phagocytosis that involves both Syk palmitoylation and desulfenylation.

## Introduction

The Syk tyrosine kinase acts downstream of several receptors that contain in their cytoplasmic tail an immunoreceptor tyrosine-based activation motif (ITAM), or a hemITAM that contains short sequences with two or one phosphorylable Tyr residues, respectively. The ITAM can also be present on a receptor-associated protein. These receptors include the Fc receptor (FcR), the T-cell and B-cell antigen receptors (TCR and BCR), C-type lectins, and the microglial triggering receptor expressed on myeloid cells 2. Syk plays a key role in adaptive immune receptor signaling, but is also involved in other cellular functions such as adhesion, osteoclast maturation, fungal pathogen and *Mycobacterium tuberculosis*

recognition, necrosis, and clearance of amyloid beta by microglia. Syk also plays a role in autoimmune diseases and hematological cancers (Mocsai et al, 2010; Ennerfelt et al, 2022; Singaram et al, 2023).

Phagocytosis by the FcγR has often been used to study the mechanism of Syk activation. Binding of IgG-opsonized particles triggers receptor activation, and Src family kinases then phosphorylate the Tyr residues within ITAMs. Syk is a 72-kD non-receptor protein tyrosine kinase that contains a kinase domain and two Src homology 2 (SH2) domains that maintain the kinase domain in an autoinhibitory conformation. Binding of the Syk SH2 domains to biphosphorylated ITAMs results in Syk activation and phosphorylation of Tyr residues, with 10 of them being autophosphorylated. Binding of the second SH2 domain to phosphatidylinositol (3,4,5) triphosphate (PIP3) is also important for Syk activation and function (Singaram et al, 2023). Syk activation takes place rapidly after phagocytosis onset, peaking after ~5 min (Raeder et al, 1999). Syk then orchestrates downstream activation pathways involving Vav family members, PLCγ, and phosphoinositide 3-kinases (PI3Ks), thereby enabling Rac1 and Cdc42 activation and actin polymerization leading to pseudopod extension and particle engulfment, enabling successful phagocytosis (Freeman & Grinstein, 2014; Mylvaganam et al, 2021). Unlike Src family kinases, Syk is strictly necessary for phagocytosis, probably because Syk can to some extent phosphorylate ITAMs (Mocsai et al, 2010).

The current Syk activation model involves conformational changes linked to phosphorylated ITAMs (p-ITAMs) and PIP3 binding and Syk phosphorylation. These changes essentially affect the interdomain B region of Syk (Mansueto et al, 2019; Singaram et al, 2023; Bradshaw et al, 2024).

Protein S-acylation (also termed palmitoylation) is a post-translational modification of cysteine residues involving the modification of the Cys-SH group by a fatty acid, most often palmitate, that becomes attached by a thioester bond to the protein. This modification is catalyzed by S-palmitoyl acyl transferases. These membrane-embedded enzymes contain a DHHC catalytic site and are often called DHHCs. In humans, among the 23 DHHCs, only 2–3 of them (depending on cell types)

[1]Institut de Recherche en Infectiologie de Montpellier, Université Montpellier, CNRS, Montpellier, France   [2]Laboratoire de Spectrométrie de Masse Bio-Organique, Université Strasbourg, CNRS, Strasbourg, France   [3]Infrastructure Nationale de Protéomique ProFI-UAR2048, Strasbourg, France   [4]Institut de Recherche en Cancérologie de Montpellier, Université Montpellier, ICM, INSERM, CNRS, Montpellier, France

Correspondence: bruno.beaumelle@irim.cnrs.fr

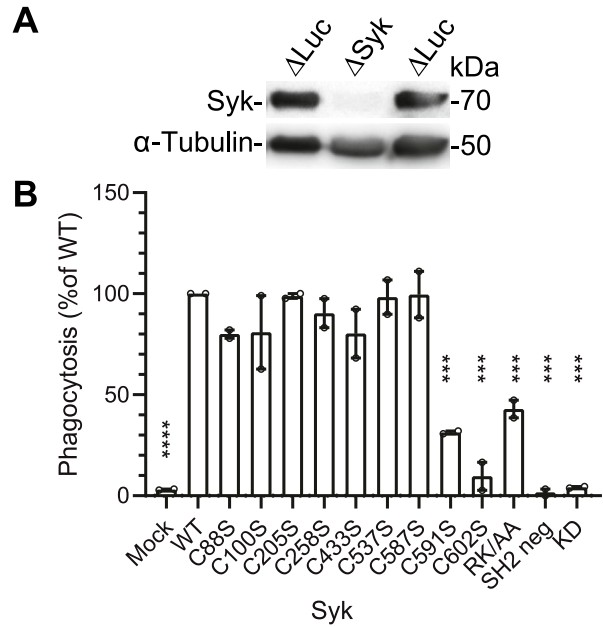

**Figure 1. Role of Syk-Cys residues in phagocytosis.**
**(A)** ΔSyk RAW 264.7 cell line does not express Syk. Western blot of cells after Syk or firefly luciferase knockout in RAW 264.7 cells using CRISPR/Cas9. **(B)** Phagocytosis efficiency of Cys-to-Ser and control mutants of mSyk after transfection of EGFP-mSyk in ΔSyk RAW 264.7 cells. Cells were allowed to phagocytose IgG-opsonized 3-μm-diameter latex beads for 15 min at 37°C, before counting beads inside and outside cells. See Fig S1 for details. Results are the mean ± SEM of two independent experiments, for which n > 100 cells were counted for each mutant. One-way ANOVA compared with WT Syk (***$P$ < 0.001). Source data are available for this figure.

## Results

### Identification of cysteine residues required for Syk to enable phagocytosis

To examine the role of Syk-Cys residues in phagocytosis, we first generated a ΔSyk RAW 264.7 mouse cell line using the CRISPR/Cas9 technique (Fig 1A). This ΔSyk RAW 264.7 cell line could not phagocytose IgG-opsonized latex beads anymore (Figs 1B and S1), confirming the key role of Syk in phagocytosis (Raeder et al, 1999). This phagocytosis defect could be restored by transient transfection with a vector expressing EGFP-mSyk WT or the Cys-to-Ser mutants of the first seven Cys residues of mSyk. Nevertheless, mutation of the two last Cys residues induced a 70–90% drop in phagocytosis efficiency compared with WT mSyk (Fig 1B). These two mutants were poorly expressed compared with the other Cys-to-Ser mutants, indicating that these Cys residues are required for Syk stability (Fig S2A). Nevertheless, the C591S and C602S mutants do not seem to be degraded by the proteasome because a treatment with MG132 failed to improve the recovery of these mutants (Fig S2B). Throughout this study (see the Materials and Methods section for details), we adapted protocols so that the weak expression of these mutants would not affect experimental results. As negative controls for phagocytosis assays, we used a catalytically dead version of mSyk (K396R [Richards et al, 1996]) and a mutant in which both SH2 domains were inactivated (R41A-R194A [Qin et al, 1998]). They were unable to support phagocytosis (Fig 1B). Mutation of mSyk-PIP3 binding site R219A-K221A (Singaram et al, 2023), termed RK/AA, inhibited phagocytosis efficiency by ~60% (Fig 1B).

### Syk is palmitoylated on its penultimate cysteine residue

Based on the weak biological activity (Fig 1B) and expression level of the corresponding C591S and C602S mutants (Fig S2), we wondered what could be the role of these two last Cys residues in the Syk biological activity. The CSS-Palm prediction software (Ren et al, 2008) suggested that mSyk could be palmitoylated on Cys591. To examine whether Syk could be palmitoylated, we first used human monocyte–derived primary macrophages (hMDMs) and the acyl-biotin exchange (ABE) technique (Chopard et al, 2018). This technique is based on the selective cleavage of the thioester bond by hydroxylamine before attaching a biotin to the unmasked SH group, enabling to purify on streptavidin-agarose the previously palmitoylated proteins that can be identified by Western blot (Wan et al, 2007). We monitored Syk palmitoylation in resting macrophages or macrophages that were allowed to phagocytose IgG-opsonized targets (sheep red blood cells [SRBCs]) for 5 min, which is the time that allows maximum Syk activation (Raeder et al, 1999). We observed that Syk was palmitoylated only upon phagocytosis (Fig 2A). Similar data were obtained using ΔSyk RAW 264.7 macrophages transfected with WT mSyk (Fig 2B). When the Cys mutants were used, we observed that the mutant of the last Syk-Cys (C602S) was also palmitoylated upon phagocytosis. The C591S mutant did not show any significant palmitoylation in the absence or presence of phagocytosis, indicating that mSyk is palmitoylated on Cys591.

localize to the plasma membrane (Chopard et al, 2018). Palmitoylation can be reversed by acyl protein esterases (APT1 or APT2) (Won et al, 2018), enabling the cell to use palmitoylation to reversibly attach proteins to membranes. A number of proteins involved in phagocytosis are already known to be S-acylated. This is the case for Src family kinases and Rac1 (Dixon et al, 2021).

The activity of Src kinases is also regulated by the sulfenylation of two cysteine residues that results in the formation of Cys-SOH. These sulfenylation events were found to modulate the conformation of the Src protein (Heppner et al, 2018). Zap70, the second member of the Syk tyrosine kinase family, is expressed in T cells and NK cells and shares ~54% of homology with Syk (Turner et al, 2000). Zap70 was found to be sulfenylated, and this modification was shown to modulate both the function and stability of the protein (Thurm et al, 2017). Zap70 was also reported to be palmitoylated (Tewari et al, 2021), but this observation could not be confirmed in a later study (Schultz et al, 2022). Our aim was to explore the eventual palmitoylation and sulfenylation of the Syk kinase, and their potential involvement in Syk activation during phagocytosis.

Here, we show that Syk is S-acylated upon phagocytosis that also seemed to trigger Syk desulfenylation. We identified the modified cysteine residues and the DHHC enzyme responsible for Syk palmitoylation. Both desulfenylation and palmitoylation are important for Syk to enable phagocytosis.

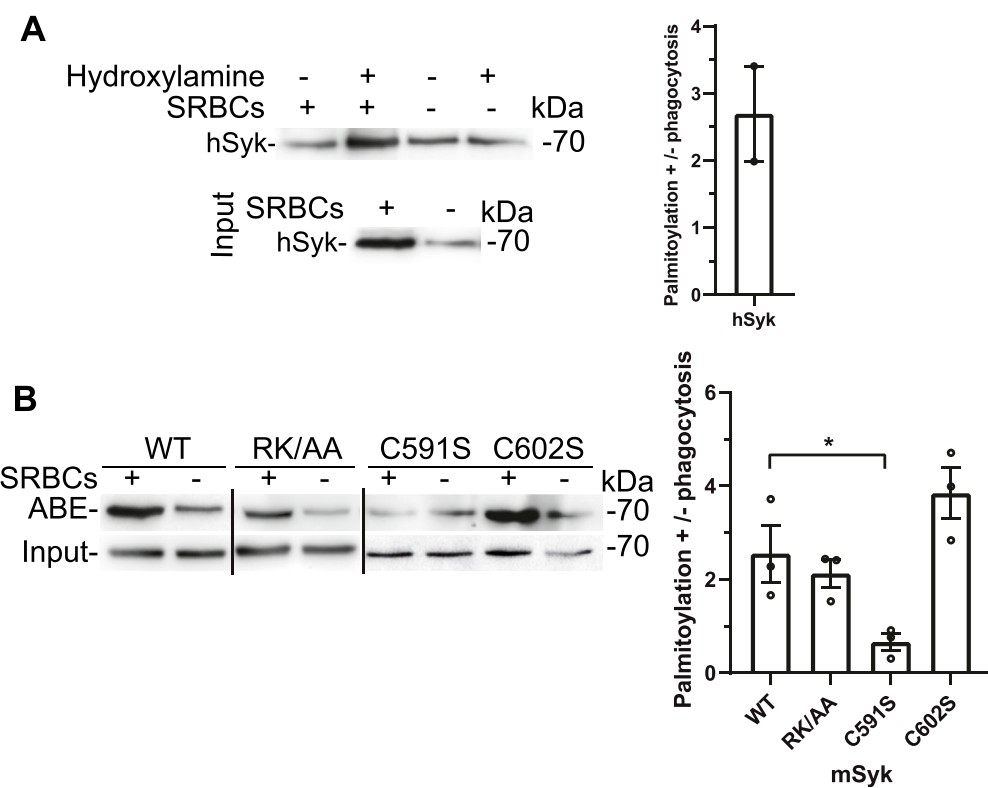

**Figure 2. Syk is palmitoylated on mSyk-Cys591.**
**(A)** hMDMs were allowed to phagocytose opsonized SRBCs as indicated, and cell lysates were treated with hydroxylamine, then HPDP-biotin to replace acyl groups by biotin, before purifying biotin-labeled proteins on streptavidin-agarose and Syk Western blot. The graph shows the quantification from n = 2 different donors. **(B)** ΔSyk RAW 264.7 cells were transfected with the indicated version of mSyk before SRBC phagocytosis as indicated, acyl-biotin exchange using hydroxylamine and Syk Western blot (ABE). Lines indicate separation between different blots. For the graph, ABE signals were normalized using Syk levels. Data are quantified from three independent experiments (mean ± SEM). One-way ANOVA (*$P < 0.05$). Source data are available for this figure.

The RK/AA mutant unable to bind PIP3 was palmitoylated, indicating that PIP3 binding is not required for palmitoylation.

### DHHC5 is responsible for Syk palmitoylation

The results showing that Syk is palmitoylated upon phagocytosis only (Fig 2) indicate that the enzyme responsible for its palmitoylation is present at the plasma membrane. When we examined the intracellular localization of the different DHHC enzymes in RAW 264.7 macrophages, we observed that in agreement with our previous study on PC12 neuroendocrine cells (Chopard et al, 2018), DHHC5 and DHHC20 only were present at the plasma membrane (Fig S3). It has been shown that DHHC20 palmitoylates and thereby regulates the activity of DHHC5 (Plain et al, 2020). We first used a mix of four siRNAs to deplete these DHHC enzymes in RAW 264.7 macrophages. The siRNAs against DHHC5 depleted ~50% of DHHC5 in RAW 264.7 cells but conversely increased by 80% their DHHC20 level (Fig 3A and B). On the other hand, siRNAs against DHHC20 reduced its RNA level by ~60% without significantly affecting the DHHC5 level. DHHC20 levels were measured using qRT–PCR because we could not detect DHHC20 by Western blots in RAW 264.7 macrophages.

When we examined the effect of these siRNAs on Syk palmitoylation, we observed that siRNAs against DHHC5 significantly decreased Syk palmitoylation, whereas siRNAs against DHHC20 had no effect (Fig 3C). Similar results were obtained when we quantified the effect of these siRNAs on phagocytosis efficiency (Fig 3D). These data are consistent with the identification of DHHC5 as the enzyme palmitoylating Syk, but do not exclude that DHHC5 could also palmitoylate other proteins essential for phagocytosis as well.

To confirm data from DHHC depletion using siRNAs, we used a DHHC overexpression approach. The expression of EGFP-mDHHC5 at a level comparable to the endogenous mDHHC5 did not significantly increase the level of Syk palmitoylation, whereas the overexpression of EGFP-mDHHC20 decreased Syk palmitoylation by threefold (Fig 3E). This is probably because EGFP-mDHHC20 overexpression decreased endogenous mDHHC5 level by ~60% (Fig 3E), a drop that is likely due to overpalmitoylation of mDHHC5, leading to its degradation (Plain et al, 2020). Hence, the ectopic expression of DHHC20 decreased DHHC5 level and Syk palmitoylation just as siRNAs targeting DHHC5.

Interestingly, upon Syk immunoprecipitation from hMDMs, we observed that DHHC5 specifically associates with Syk upon phagocytosis (Fig 4A). Because Syk is recruited to the phagocytic cup upon phagocytosis (see below), DHHC5 should also concentrate at this level. When WT RAW 264.7 macrophages were transfected with EGFP-mDHHC5 or EGFP-mDHHC20, we observed that DHHC5 was recruited to the phagocytic cup, whereas this was not the case for DHHC20 (Figs 4B and S4). Quantification of line plots shown in Fig 4B indicated that DHHC5 was ~2-fold more concentrated at the cup compared with control areas of the plasma membrane (Fig 4C). When this experiment was performed using ΔSyk cells that are unable to perform phagocytosis (Fig 1B), no significant recruitment of DHHC5 at the cup was observed. Altogether, these results show that DHHC5 is responsible for Syk

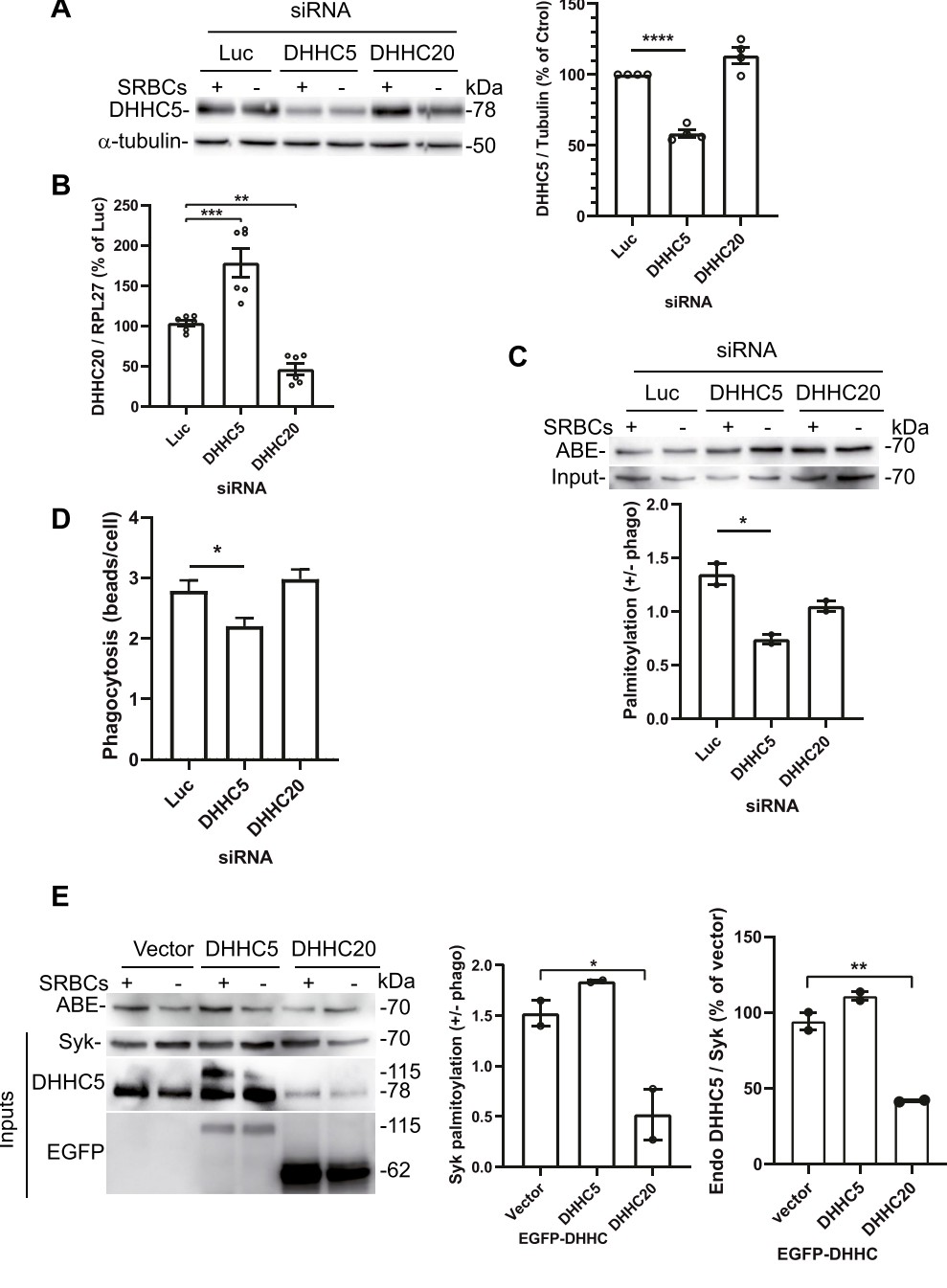

**Figure 3.   Syk is palmitoylated by DHHC5.**
**(A)** RAW 264.7 cells were transfected with the indicated siRNA before phagocytosis of opsonized SRBCs for 5 min, cell lysis, and Western blot against DHHC5 and tubulin. The graph shows the quantification (mean ± SEM) from two independent experiments including ± phagocytosis data for each of them.
**(B)** RNA was extracted from transfected cells before RT–qPCR for DHHC20 and RLP27. Data are means ± SEM (two experiments performed in triplicate).
**(C)** siRNA-transfected RAW 264.7 cells were allowed to phagocytose opsonized SRBCs before acyl-biotin exchange (ABE), purification on streptavidin-agarose, and Syk Western blot. The graph shows the quantification (n = 2 independent experiments).
**(D)** Depletion in DHHC5 inhibits phagocytosis. RAW 264.7 cells were transfected with the indicated siRNAs before assaying phagocytosis efficiency (means ± SEM, n > 178 transfected cells/condition). **(E)** Effect of DHHC5 or DHHC20 overexpression on Syk palmitoylation. RAW 264.7 cells were transfected with the indicated EGFP-mDHHC before phagocytosis of opsonized SRBCs, cell lysis, and Syk palmitoylation assay using ABE. EGFP-mDHHC5 was detected at 115 kD using anti-DHHC5 and anti-EGFP, whereas endogenous mDHHC5 is at 78 kD. EGFP-mDHHC20 (62 kD) could only be detected using anti-EGFP. The graphs show the efficiency of Syk palmitoylation (+/− phagocytosis) and the effect of DHHC overexpression on the endogenous DHHC5 level for n = 2 independent experiments. One-way ANOVA compared with vector or Luc siRNA ($*P < 0.05$; $**P < 0.01$; $***P < 0.001$; $****P < 0.0001$).
Source data are available for this figure.

palmitoylation upon phagocytosis. They also indicate that Syk regulates its palmitoylation through the recruitment of DHHC5 at the phagocytic cup upon phagocytosis induction.

### Role of Syk-Cys602 in the sulfenylation process and the regulation of Syk structure

Syk is palmitoylated on its penultimate Cys (C591 for mSyk), and this modification is important for Syk to sustain phagocytosis (Fig 1B), but the role of the last Cys (Cys602 for mSyk) that is strictly

needed for phagocytosis remained unclear. Syk contains in its C-terminal sequence a redox-sensitive motif previously identified in several non-receptor tyrosine kinases such as Src and Zap70 (Fig S5). The Cys within this motif was found to be sulfenylated in the Syk-family kinase Zap70 (Thurm et al, 2017).

We first examined whether Syk is sulfenylated using hMDMs and the specific reagent for Cys-SOH termed DCP-Bio1 (Hourihan et al, 2016) that enables to label sulfenylated Cys with biotin. Syk was purified by immunoprecipitation from DCP-Bio1–treated extracts, and biotin staining revealed a large number of bands, including the

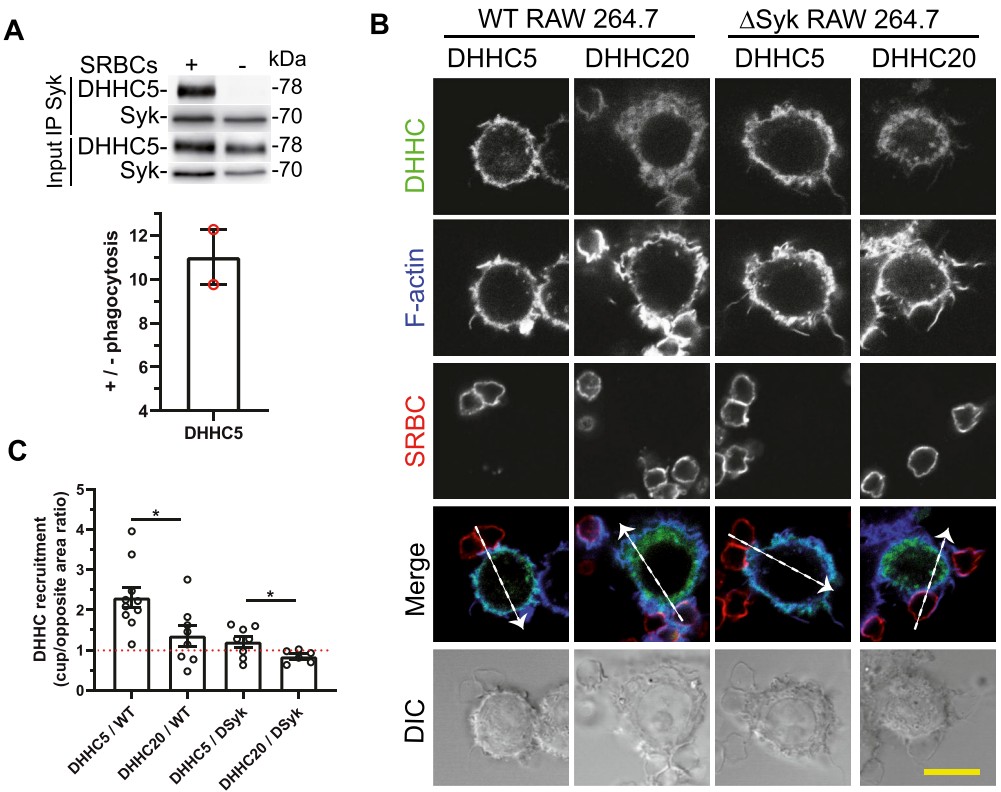

**Figure 4. Syk interacts with DHHC5.**
**(A)** DHHC5 interacts with Syk upon phagocytosis. hMDMs were allowed to phagocytose opsonized SRBCs, before Syk immunoprecipitation and Syk and DHHC5 Western blots. The graph shows the quantification +/− phagocytosis (mean ± SEM for two different donors). **(B)** DHHC5 but not DHHC20 is recruited to the phagocytic cup. RAW 264.7 macrophages (WT or ΔSyk) were transfected with EGFP-DHHC5 or EGFP-DHHC20 before adding opsonized SRBCs for 5 min, fixation, F-actin and SRBC labeling, and confocal microscopy. Bar, 10 μm. Line plots (width: 10 pixels) showed on the merged images enable to follow the signal accumulation at the phagocytic cup. Representative plots are shown in Fig S4. **(C)** Graph shows the ratio (mean ± SEM) of the fluorescence peaks at the cup to those on the opposite side of the cell, from n = 6–10 cells. One-way ANOVA was used for comparison of DHHC5 with DHHC20 accumulation (*$P < 0.05$). Source data are available for this figure.

one corresponding to hSyk. Quantification indicated that Syk was sulfenylated and that its sulfenylation level was decreased by 20% upon phagocytosis induction (Fig 5A). To examine the specificity of DCP-Bio1 Syk labeling, we treated RAW 264.7 macrophages cells with N-acetyl cysteine (NAC) to prevent Cys oxidation (Krasnowska et al, 2008). NAC treatment decreased by more than 55% Syk labeling by DCP-Bio1 (Fig 5B). The same inhibition of 55% was observed for sulfenylated proteins that co-immunoprecipitate with Syk, indicating that DCP-Bio1 labeling is specific for sulfenylated proteins. This incomplete inhibition of protein sulfenylation by NAC could be due to the use of a suboptimal NAC concentration (Krasnowska et al, 2008). No sulfenylated proteins were observed after Syk immunoprecipitation from ΔSyk RAW 264.7 cells treated with DCP-Bio1 (Fig 5B). When we used ΔSyk RAW 264.7 macrophages transfected with mSyk, we observed that Syk was desulfenylated upon phagocytosis. DCP-Bio1 labeling in RAW 264.7 cells produced faint bands, but quantification of four independent experiments suggested that, unlike WT mSyk, neither Syk-RK/AA, nor C591S or C602S was desulfenylated upon phagocytosis (Fig 5C). These experiments indicated that Syk is desulfenylated upon phagocytosis and that PIP3 binding and palmitoylation are important for Syk desulfenylation. The identification of Cys602 as an oxidized residue in mSyk was confirmed by proteomic analysis of ΔSyk RAW 264.7 macrophages transfected with FLAG-Syk. This approach enabled efficient Syk immunoprecipitation using anti-FLAG antibodies, resulting in 88% of protein sequence coverage by mass spectrometric analysis, including peptides of interest such as the 594–619 peptide. Fragmentation of this peptide indicated that

Cys602 was sulfinylated (Cys602-SO$_2$H) and Met599 oxidized (Fig S6). This Cys602 sulfinylation likely results from the oxidation of the sulfenylated Cys (Cys602-SOH) during sample purification and processing (Heppner et al, 2018). Identification of Cys602 as Syk sulfenylated residue is also consistent with the localization of Cys602 within Syk redox-sensitive motif (Fig S5).

To examine the effect of sulfenylation on the Syk structure, we performed molecular dynamics simulations. We observed that despite the fact that the sulfenylated Cys (C608 for hSyk) is located in a different region of the protein (Fig 6A), the sulfenylation of Cys608 decreases the mobility of a flexible loop (residues 267–325) within the interdomain region (Fig 6B). The replacement of the sulfenylable Cys by a Ser (hSyk-C608S, equivalent to the mSyk-C602S) further impairs the mobility of this loop. Moreover, for the WT Syk protein, the loop movements were more oriented toward the outside of protein compared with the sulfenylated hSyk-C608-OH protein or hSyk-C608S (Fig 6C). This enhanced flexibility of the WT Syk loop could favor the autophosphorylation of the single Tyr residue of this loop, Tyr296, which is the first phosphorylated Syk-Tyr upon activation. The role of this residue in Syk activation is not clear because Syk-Y296F shows native kinase activity, but weakly phosphorylates the key Tyr348/352 residues (Mansueto et al, 2019). The simulation time we used (500 ns) was insufficient to observe structuration of interdomain B. Regarding the interaction of the sulfenylated Cys, we observed during molecular dynamics simulations that in WT hSyk, Cys608 is stably connected to Leu604 within the same α-helix by a hydrogen bond (Fig S7). Nevertheless, sulfenylation somehow destabilizes this interaction

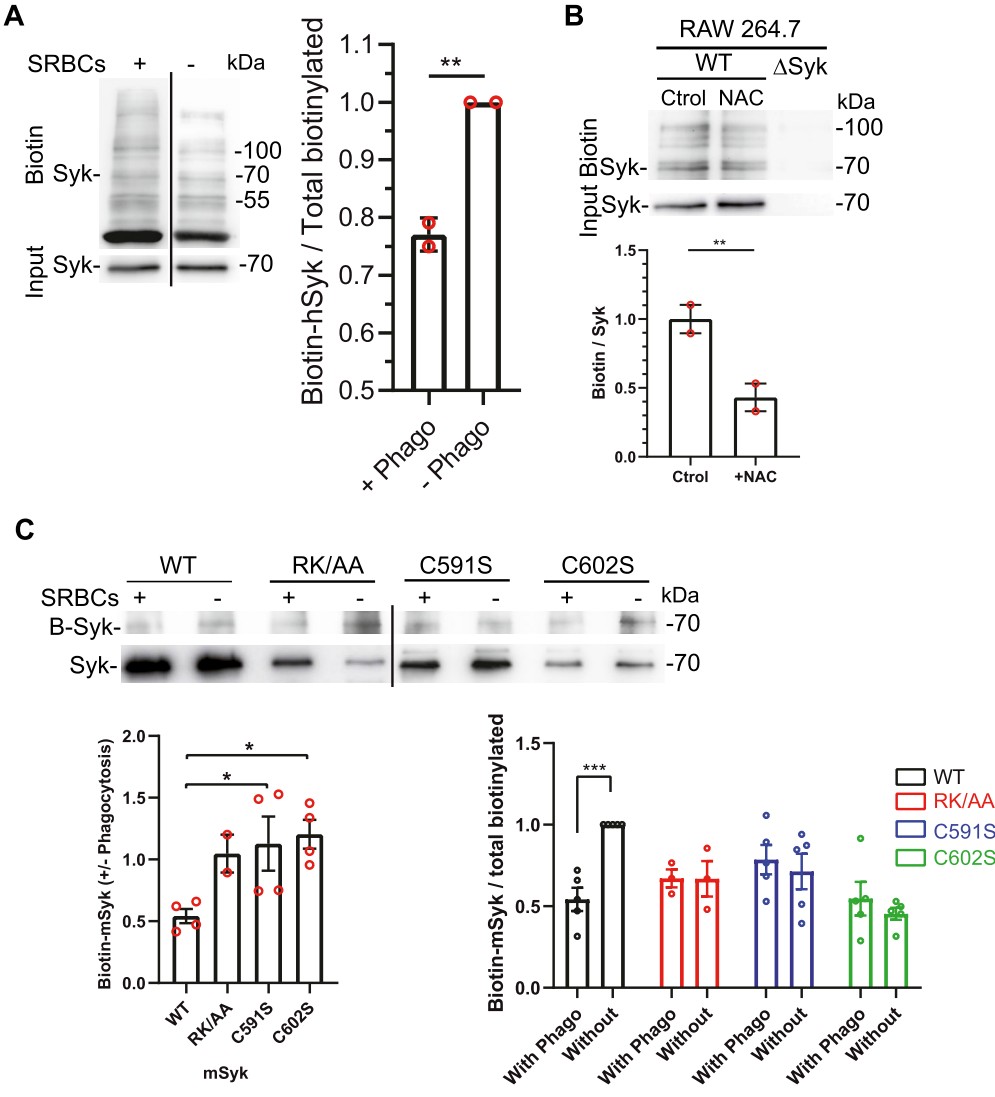

**Figure 5. Syk is desulfenylated upon phagocytosis induction.**
**(A)** hMDMs were allowed to phagocytose opsonized SRBCs as indicated, and cell lysates were treated with DCP-Bio1 to attach biotin to sulfenylated Cys. After Syk immunoprecipitation and Western blot, biotin was detected using ExtrAvidin-peroxidase. The line indicates separation between different lanes of the same blot. The graph shows the quantification of biotin-Syk (mean ± SEM, n = 2 different donors). Unpaired $t$ test (**$P < 0.01$). **(B)** DCP-Bio1 labeling is inhibited by N-acetyl Cys (NAC). RAW 264.7 macrophages (WT or ΔSyk) were treated with NAC as indicated before DCP-Bio1 treatment of cell lysates, Syk immunoprecipitation, and biotin staining. The graph shows the quantification (mean ± SEM) from n = 2 experiments. Paired $t$ test (**$P < 0.01$). **(C)** mSyk is sulfenylated on Cys602. ΔSyk RAW 264.7 cells were transfected with the indicated mSyk mutant before phagocytosis of opsonized SRBCs. Cell lysates were treated with DCP-Bio1 before mSyk immunoprecipitation, Western blot, and biotin staining. The line indicates separation between different blots. The graph shows the quantifications of biotin-Syk (mean ± SEM, n = 4 independent experiments). One-way or two-way ANOVA (*$P < 0.05$; ***$P < 0.001$). Source data are available for this figure.

and sulfenylated Cys608 sometimes (~20% of the simulation time; Fig S7) establish H-bonding with Val555 that is within another α-helix. This interaction might explain the long-distance effect on the flexible interdomain. Altogether, these molecular dynamics experiments indicated that Syk sulfenylation induces a more stabilized and packed Syk structure, whereas desulfenylation allows molecule opening at the level of both the kinase domain and the interdomain B.

## Syk is required for the transient production of $H_2O_2$ at the phagocytic cup

It is well established that $H_2O_2$ is produced upon phagocytosis. Nevertheless, $H_2O_2$ production was most often measured after several minutes of phagocytosis (Root et al, 1975; Ueyama et al, 2004; Abo et al, 2014), and the production of $H_2O_2$ at the onset of phagocytosis remained to be examined. The FRET probe HyPer7 enables ultrasensitive, specific, and real-time detection

of $H_2O_2$ in living cells. We used the HyPer7-LifeAct chimera that binds to actin filaments (Pak et al, 2020) and localized to the cortical actin cytoskeleton in macrophages (Fig 7). $H_2O_2$ is an uncharged molecule that easily crosses membranes (Smolyarova et al, 2022), and HyPer7-LifeAct should thus enable to monitor $H_2O_2$ production at the phagocytic cup of WT RAW 264.7 cells. Using live confocal microscopy, we observed that $H_2O_2$ production is initiated at the cup after 160 s of phagocytosis, and vanished after 200 s when the phagosome is formed, as shown in Fig 7 that presents images selected from Video 1, which lasts 10 min. When ΔSyk cells were used, neither $H_2O_2$ production nor phagocytosis was observed, and their $H_2O_2$ production was lower compared with WT cells (Fig 7 and Video 2). $H_2O_2$ is known to activate Syk, and this activation begins after 2 min of exposure, then drops after 5 min (Qin et al, 1998). The sulfenylation/desulfenylation cycle of Syk could thus be linked to the transient $H_2O_2$ production that occurs at the cup.

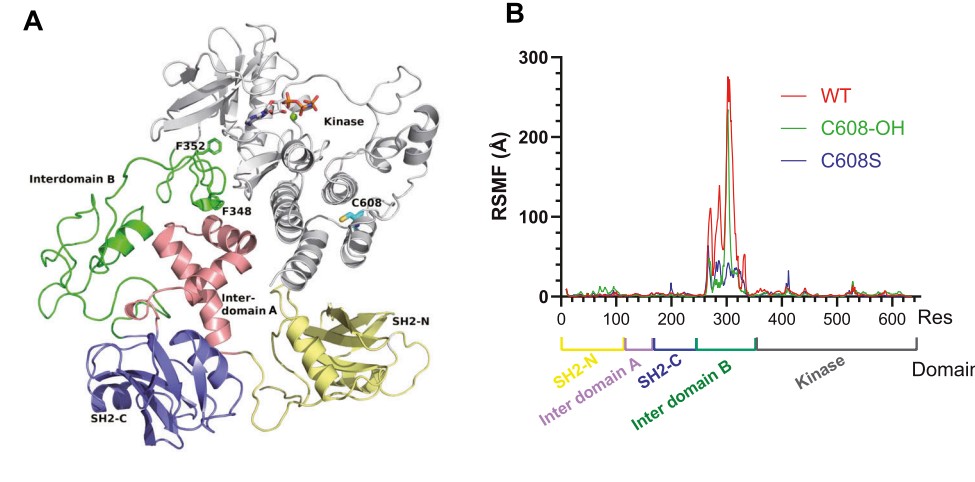

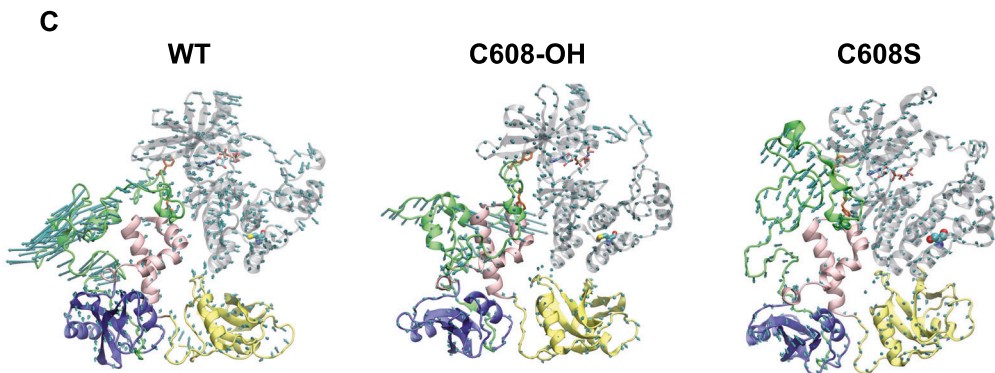

**Figure 6. Molecular dynamics studies indicate that Syk desulfenylation induces a higher mobility and exposure of a loop that includes the 263–335 residues.**
**(A)** 3D model of hSyk based on the PDB-4FL3 crystal structure. The localization of the sulfenylated hSyk-Cys608 (Cys602 for mSyk) is indicated. Y348 and Y352 are both replaced by Phe in the 4FL3 construct, and these residues are depicted in sticks. The different domains of Syk are color-coded with SH2-N (yellow), SH2-C (blue), kinase (gray), interdomain A (pink), and the modeled interdomain B (green). **(B)** Mobility of the protein backbone by computing the atomic fluctuations (RMSF, Å) highlighting the flexible loop motion for WT Syk, sulfenylated Syk (C608-OH), and C608S mutant. **(C)** 3D representations of the motions measured in panel (B) for WT, sulfenylated, and C608S Syk. The arrows indicate the amplitude of the motions. Note that for the WT, the loop is moving toward the outside of the protein, whereas for sulfenylated hSyk, movements are smaller and directed toward the center of the molecule. These motions are dramatically reduced for the C608S mutant.
Source data are available for this figure.

## Sulfenylation, palmitoylation, and binding to PIP3 are important for Syk phosphorylation and activation

We then monitored the consequences of sulfenylation, palmitoylation, and binding to PIP3 on Syk activation by following its phosphorylation on two key Tyr residues (525/526 for hSyk and 519/520 for mSyk) within Syk activation loop, in the kinase domain. Their phosphorylation was found to be a prerequisite for Syk activation (Zhang et al, 2000). Using the corresponding mSyk mutants, we found that catalytic activity, SH2 binding, desulfenylation, palmitoylation, and PIP3 binding are all required for Syk activation when followed using the phosphorylation of these key Tyr residues (Fig 8).

## Syk sulfenylation/desulfenylation is important for Syk catalytic activity

We then examined the effects of sulfenylation, palmitoylation, and binding to PIP3 on Syk catalytic activity. To this end, we used an in cellulo FRET assay (Xiang et al, 2011). We first monitored the kinetics of Syk activation after the onset of phagocytosis. In agreement with the previously reported kinetics of Syk phosphorylation (Raeder et al, 1999), we observed that Syk catalytic activity is maximal after 5 min of phagocytosis (Fig 9A). When we assessed the catalytic activity of the different Syk mutants, we found that the C602S was

inactive, whereas Syk-591S and Syk-RK/AA were catalytically active (Figs 9B and S8). These results thus indicated that sulfenylation/desulfenylation is important for Syk catalytic activity, whereas PIP3 binding and palmitoylation are not necessary. Control mutants showed that Syk catalytic site and SH2 binding are both critical for Syk kinase activity in cellulo.

## Syk palmitoylation is important for its localization at the phagocytic cup

Because palmitoylation usually stabilizes the association of proteins with membranes (Ko & Dixon, 2018), we assessed whether Syk palmitoylation affected its localization at the phagocytic cup. To this end, we used RAW 264.7 macrophages transfected with EGFP-Syk. As previously observed in B cells (Ma et al, 2001), Syk is essentially cytosolic and recruited to the cell membrane upon stimulation (Fig 10A). We followed Syk accumulation at the phagocytic cup using line plots across the cell and compared Syk accumulation at the cup with that on the opposite region of the plasma membrane (Figs 10B and S9). Statistical analysis showed that Syk was ~4-fold more concentrated at the cup compared with control areas of the plasma membrane (Fig 10B). Non-palmitoylable Syk (C591S) was only 1.9-fold concentrated at the cup, whereas non-sulfenylable Syk (C602S) and Syk-RK/AA unable to bind PIP3 behaved as WT Syk. Control Syk mutants with

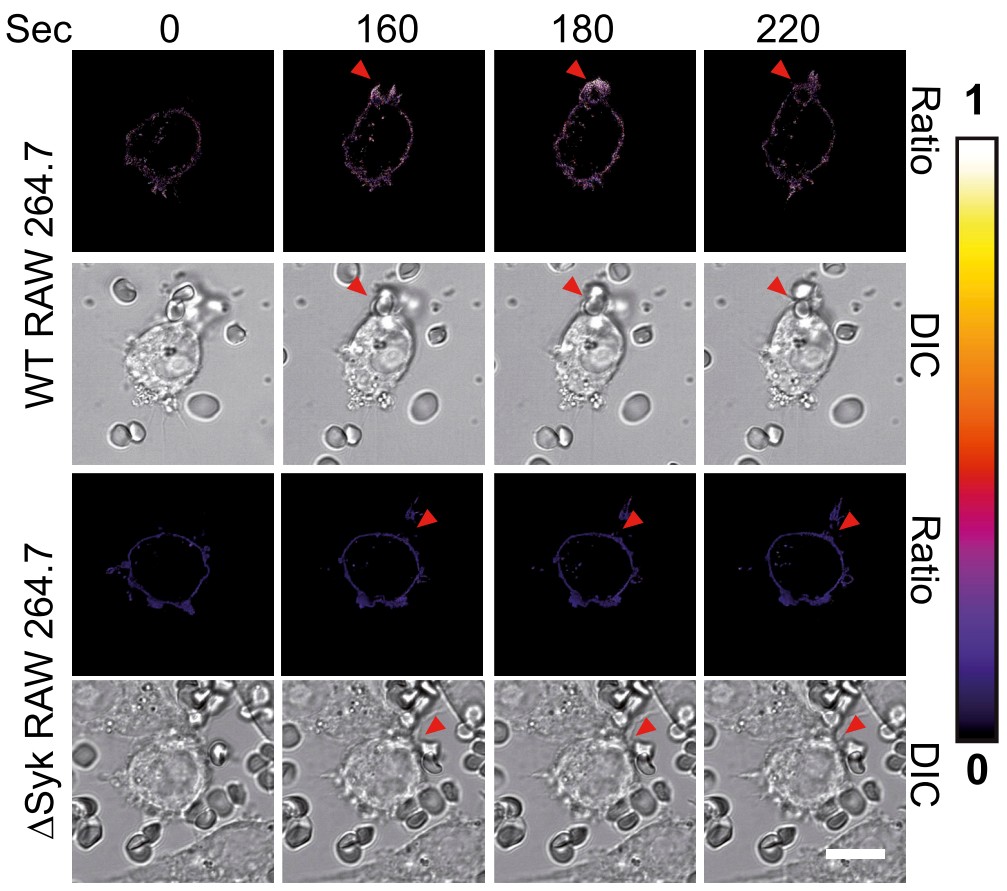

**Figure 7. H₂O₂ production at the phagocytic cup.**
RAW 264.7 macrophages (WT or ΔSyk) were transfected with pLifeAct-HyPer7. After 18 h, they were placed on the stage of an inverted confocal microscope at 37°C before adding opsonized SRBCs. When an SRBC contacted a macrophage, images were recorded at 530 nm every 20 s for 10 min. Images present the ratio of intensities (excitation 490 nm)/(excitation 420 nm) that is proportional to H₂O₂ production, as indicated on the color scale (0–1). The ratio for ΔSyk cells is always lower compared with WT cells. Bar, 10 μm.

inactivated SH2 domains or catalytic site did not accumulate at the cup. These results indicate that Syk catalytic activity, p-ITAM binding, and palmitoylation are essential to ensure Syk localization at the phagocytic cup, whereas sulfenylation/desulfenylation and PIP3 binding do not seem to be required.

### Syk palmitoylation and catalytic activity are important for DHHC5 recruitment at the phagocytic cup

Syk mutants that did not accumulate at the phagocytic cup failed to recruit DHHC5 at this level (Figs 10C and S10). This is the case of the non-palmitoylable mutant (C591S) and control mutants with inactivated SH2 or catalytic activity. The absence of significant DHHC5 recruitment by mutants deficient in catalytic activity (KD and C602S) indicates that Syk kinase activity is required for the recruitment of DHHC5 at the phagocytic cup. Hence, both Syk recruitment at the cup and Syk catalytic activity are necessary for DHHC5 recruitment at this level.

### Syk palmitoylation regulates the recruitment of Cdc42 but not Rac1 at the phagocytic cup

The Rho-family GTPases Rac1 and Cdc42 are key regulators of the actin polymerization process that enables pseudopod extension during phagocytosis (Freeman & Grinstein, 2014; Mylvaganam et al,

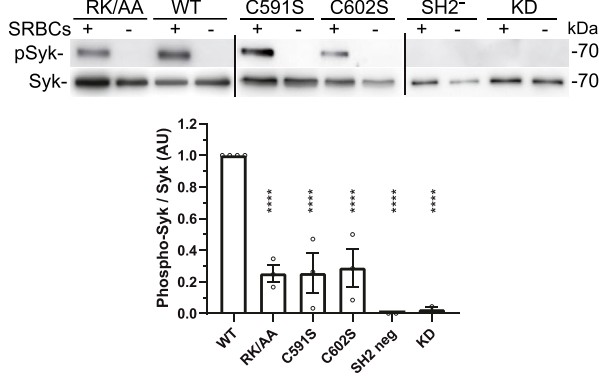

**Figure 8. Sulfenylation, palmitoylation, and PIP3 binding of Syk are required for efficient Syk phosphorylation upon phagocytosis.**
ΔSyk RAW 264.7 cells were transfected with the indicated mSyk mutant before phagocytosis of opsonized SRBCs for 5 min, SDS–PAGE, and Western blot using an anti-phosphoTyr-mSyk 519/520 (equivalent to Tyr525/526 of hSyk). Lines indicate separation between different blots. The graph shows the quantification of n = 4 independent experiments (mean ± SEM). One-way ANOVA compared with WT (****$P < 0.0001$).
Source data are available for this figure.

2021). We examined the role of Syk modifications on the recruitment at the cup of these two GTPases. WT Syk enabled Rac1 recruitment at the cup, with a threefold average enrichment

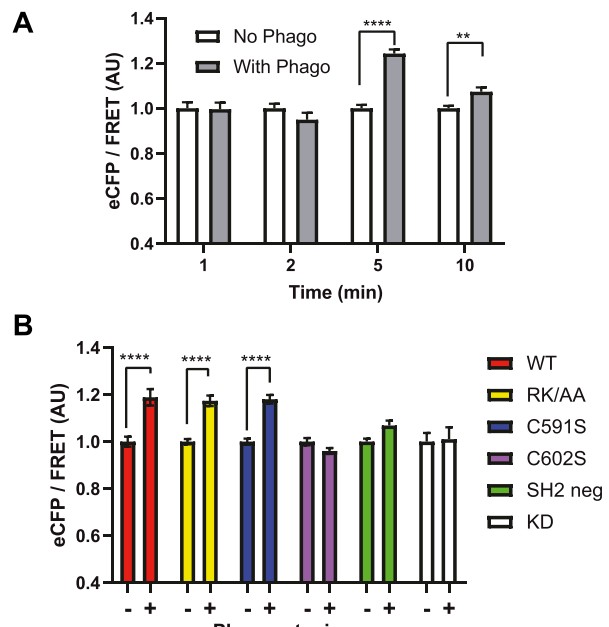

**Figure 9. Syk sulfenylation/desulfenylation but not palmitoylation or PIP3 binding is required for Syk catalytic activity.**
**(A)** RAW 264.7 cells were transfected with the FRET Syk biosensor before phagocytosis of opsonized SRBCs for the indicated time. After fixation, cells (n > 50) were imaged using a confocal microscope to monitor ECFP and FRET fluorescence. Results were normalized using 1-min values. **(B)** ΔSyk RAW 264.7 cells were transfected with the FRET Syk biosensor and the indicated mSyk mutant before phagocytosis of opsonized SRBCs for 5 min. After fixation, cells (n > 30) were imaged using a confocal microscope. Representative images are shown in Fig S8. Two-way ANOVA (**$P < 0.01$; ****$P < 0.0001$). Source data are available for this figure.

(Figs 11A and C and S11). No GTPase recruitment was observed for C602S, SH2-deficient, and kinase-dead Syk mutants, indicating that sulfenylation/desulfenylation, catalytic activity, and p-ITAM binding are required for Syk to trigger Rac1 recruitment at the cup. The Syk-RK/AA mutant unable to bind PIP3 behaved as the WT. When we examined the consequences of Syk mutants on the recruitment of Cdc42, similar data were obtained (Figs 11B and C and S12), except for the C591S mutant that enabled the recruitment of Rac1 (Figs 11A and C and S11) but not Cdc42 (Figs 11B and C and S12). These results indicate that Syk palmitoylation is required for the Cdc42 but not Rac1 recruitment at the phagocytic cup.

### Syk sulfenylation/desulfenylation and palmitoylation are important for PIP3 and DAG production at the phagocytic cup

Syk is required for PIP3 and DAG production at the phagocytic cup (Freeman & Grinstein, 2014; Mylvaganam et al, 2021). We followed the production of these second messengers using fluorescent chimeras that specifically bind these lipids. The different Syk mutants similarly affected the production of both PIP3 and DAG at the cup (Figs 12A and S13). Although sulfenylation/desulfenylation, SH2-deficient, and kinase-dead mutants did not allow significant production of these second messengers, the non-palmitoylable Syk mutant was ~50% less efficient than WT Syk in inducing

PIP3 and DAG production. The RK/AA mutation had no effect, indicating that Syk binding to PIP3 is not required for the production of these messengers.

### All Syk modifications are required to allow actin polymerization at the phagocytic cup

F-actin polymerization at the phagocytic cup is driving pseudopod extension and phagocytosis (Freeman & Grinstein, 2014; Mylvaganam et al, 2021). We found that all Syk mutants were affected in their capacity to initiate F-actin polymerization at the cup (Fig 12B), that essentially corresponds to their ability to sustain phagocytosis (Fig 1B). Although the PIP3 binding–deficient mutant showed an intermediate phenotype, the mutants deficient in sulfenylation/desulfenylation, p-ITAM binding, or catalytic activity prevented almost entirely F-actin concentration at the cup and phagocytosis. The non-palmitoylable mutant, despite the fact that it allowed only minute amounts of actin accumulation at the cup (Fig 12B), enabled significant phagocytosis (~30% of WT, Fig 1B).

### An updated model for Syk activation during phagocytosis

Based on our results (summarized in Table 1), we propose the following model for the activation of Syk during phagocytosis (graphical abstract). The first step consists of the Syk recruitment at the phagocytic cup by p-ITAMs. This initial membrane targeting allows Syk to encounter and interact with DHHC5, which is a membrane protein (Chopard et al, 2018). Non-palmitoylable Syk does not stay at the cup (Fig 10) and failed to recruit DHHC5 (Fig 10C), but is nevertheless able to initiate Rac1 recruitment (Fig 11A). Syk palmitoylation secures its association with the cup and allows more PIP3 production and consequently more significant PIP3 binding. Palmitoylation is also required for Cdc42 recruitment at the cup.

Results obtained using the non-palmitoylable Syk mutant indicate that desulfenylation takes place after palmitoylation and Syk recruitment at the cup and that sulfenylation/desulfenylation is important for the Syk catalytic activity. Cdc42 recruitment at the cup involves both Syk palmitoylation and desulfenylation. The Syk mutant that is unable to bind PIP3 is not affected in the early steps of mSyk activation (Syk, DHHC5, Rac1, and Cdc42 recruitment at the cup, PIP3 and DAG production), but it nevertheless failed to be phosphorylated on the key 519/520 Tyr residues involved in mSyk activation (Fig 8) and to initiate F-actin polymerization (Fig 12B), indicating a crucial role of PIP3 binding in the late stages of Syk activation.

## Discussion

We here provide evidence that Syk undergoes two previously unknown posttranslational modifications, palmitoylation and sulfenylation/desulfenylation. Palmitoylation and desulfenylation were found to specifically take place during Syk activation upon FcγR-mediated phagocytosis in macrophages. A proteomic study of S-acylated proteins of RAW 264.7 macrophages (Merrick et al, 2011)

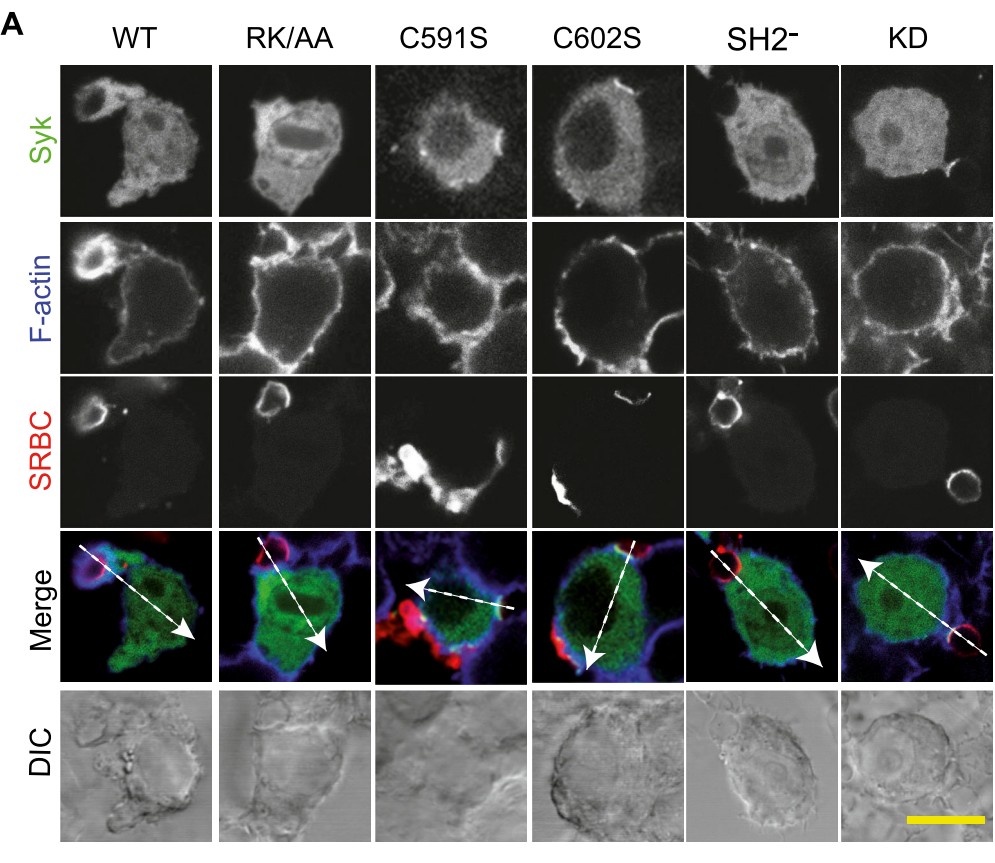

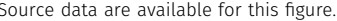

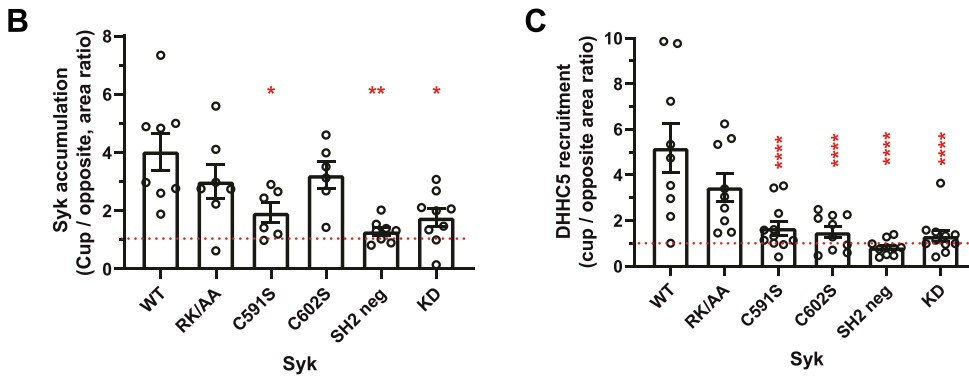

**Figure 10. Syk palmitoylation is required for the recruitment of Syk and DHHC5 at the phagocytic cup.**
**(A)** ΔSyk RAW 264.7 cells were transfected with the indicated EGFP-mSyk mutant before phagocytosis of opsonized SRBCs for 5 min. Cells were then fixed and stained with fluorescent anti-opsonizing antibody and phalloidin before confocal microscopy. Bar, 10 μm. Line plots (width: 10 pixels) showed on the merged images enabled to follow signal accumulation at the phagocytic cup. Representative plots are shown in Fig S9. **(B)** Graph shows the ratio (mean ± SEM) of the fluorescence peaks at the cup to those on the opposite side of the cell, from n = 6–9 cells. **(C)** ΔSyk RAW 264.7 macrophages were transfected with EGFP-DHHC5 and the indicated Syk mutant before SRBC phagocytosis, fixation, F-actin, and SRBC labeling, and confocal microscopy. Representative images are in Fig S10. The graph shows the ratio (mean ± SEM) of the fluorescence peaks at the cup to those on the opposite side of the cell, from n = 10–12 cells. One-way ANOVA compared with WT Syk (*$P$ < 0.05; **$P$ < 0.01; ****$P$ < 0.0001).
Source data are available for this figure.

did not identify Syk because the macrophages were not stimulated. We identified mSyk-C591 (equivalent to hSyk-C597) as the palmitoylated residue. This residue is equivalent to Zap70-C564 that was suggested to be palmitoylated upon TCR stimulation (Tewari et al, 2021), a result that could not be confirmed in a later study (Schultz et al, 2022). Using both siRNA-based and overexpression approaches, we observed that Syk palmitoylation is performed by DHHC5 that could be co-immunoprecipitated with Syk upon phagocytosis only. We showed that Syk palmitoylation by DHHC5 is regulated by DHHC20 that regulates DHHC5 intracellular level and palmitoylation (Plain et al, 2020). Palmitoylation is known to stabilize the association of cytosolic proteins once targeted to the membrane so that they can meet the appropriate DHHC enzyme

(Chopard et al, 2018). Interestingly, we observed that Syk, after its recruitment at the phagocytic cup and if catalytically active, triggers the accumulation of DHHC5 at the cup (Figs 4B and 10C), thereby favoring Syk palmitoylation. The regulation of Syk palmitoylation is likely complex because we observed that DHHC5 levels are regulated by DHHC20 in RAW 264.7 cells (Fig 3), confirming previous observations (Plain et al, 2020). Unpalmitoylable Syk seems to be unable to recruit DHHC5 at the cup (Fig 10C).

Syk recruitment at the cup requires its binding to p-ITAMs (Fig 10A) that enables its palmitoylation so that Syk remains at the cup and efficiently initiates downstream signaling such as Cdc42 recruitment, PIP3 and DAG production, and, eventually,

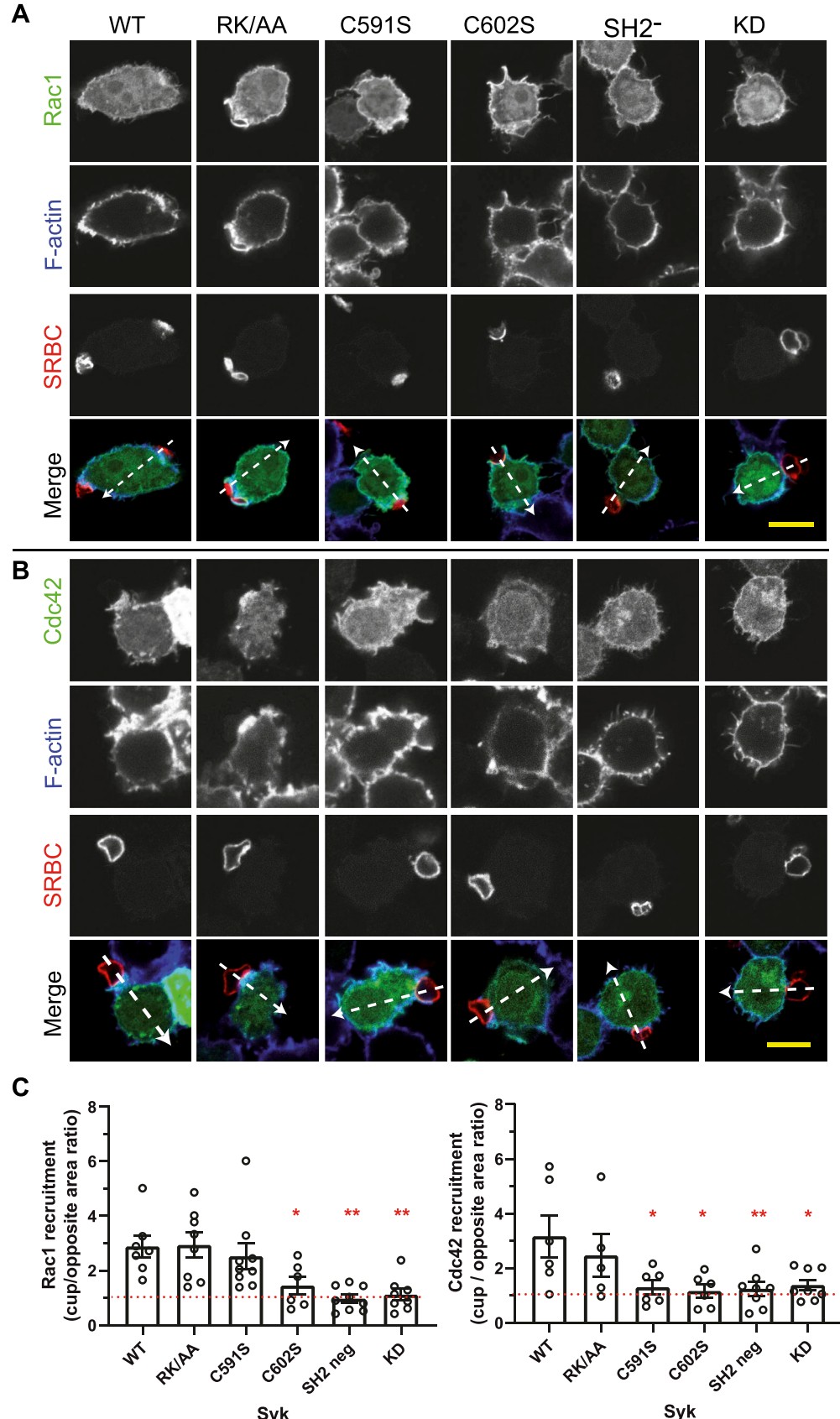

**Figure 11. Syk palmitoylation is required for the recruitment of Cdc42 but not of Rac1 at the phagocytic cup.**

**(A, B)** ΔSyk RAW 264.7 cells were transfected with vectors encoding for (A) EGFP-Rac1 or (B) EGFP-Cdc42, and the indicated FLAG-mSyk mutant before phagocytosis of opsonized SRBCs for 5 min. Cells were then fixed and stained with fluorescent anti-opsonizing antibody and phalloidin. Line plots (width: 10 pixels) showed on the merged images enabled to follow signal accumulation in the phagocytic cup. Representative plots are shown in Fig S11 (for Rac1) and Fig S12 (for Cdc42). Bars, 10 μm. **(C)** Graphs show the ratio (mean ± SEM) of the areas of the fluorescence peaks at the cup to those on the opposite side of the cell, from n = 6–9 cells. One-way ANOVA compared with WT (*$P < 0.05$; **$P < 0.01$). Source data are available for this figure.

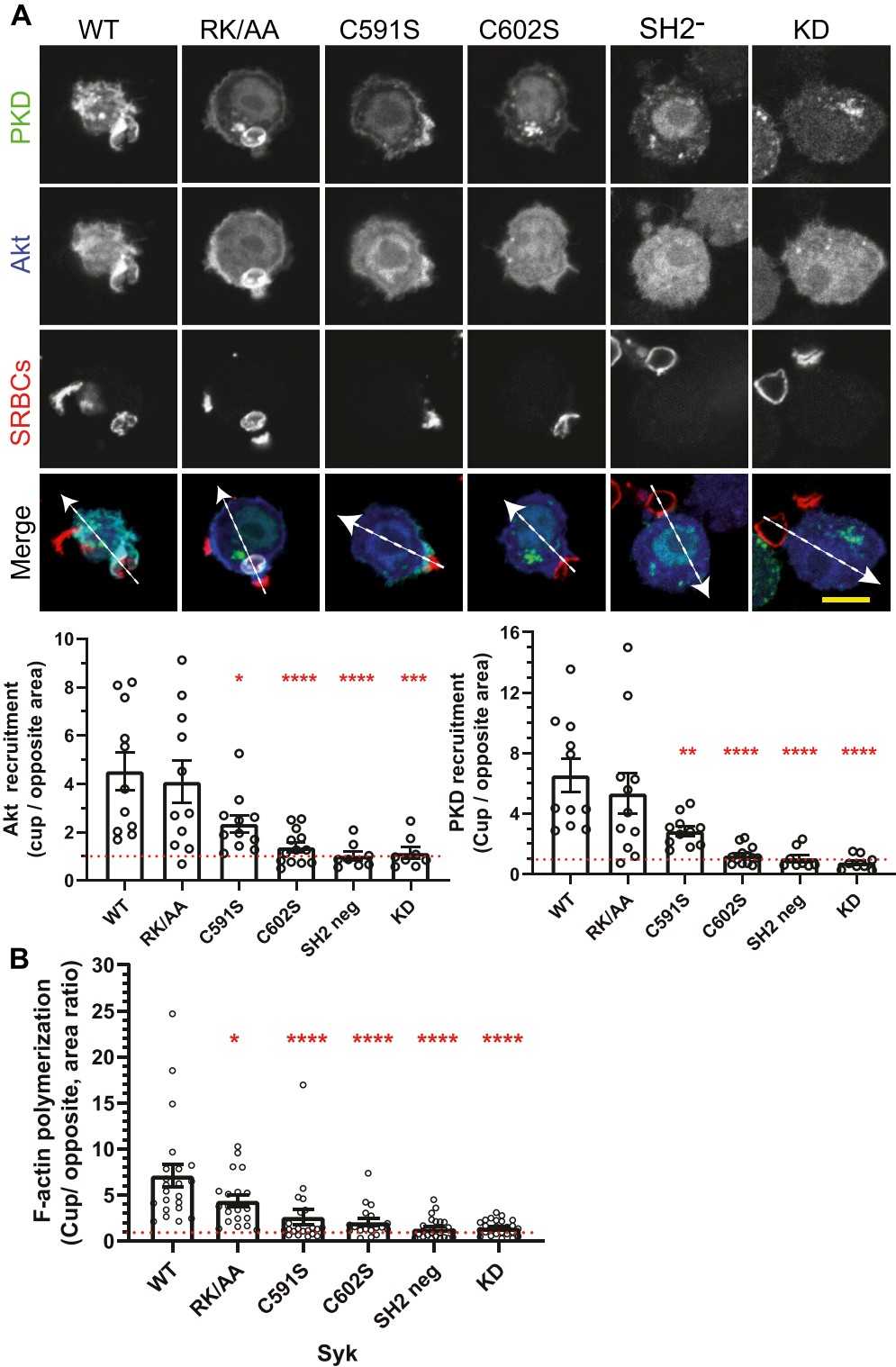

**Figure 12. Syk sulfenylation and palmitoylation are required for PIP3 and DAG production, and for F-actin polymerization at the phagocytic cup.**
**(A)** ΔSyk RAW 264.7 cells were transfected with vectors encoding for EGFP-PKD-C1ab (a DAG probe), mCherry-Akt (a PIP3 probe), and the indicated FLAG-mSyk mutant as indicated before phagocytosis of opsonized SRBCs for 5 min. Cells were then fixed and stained with fluorescent anti-opsonizing antibody and phalloidin. Line plots (width: 10 pixels) showed on the merged images enabled to follow signal accumulation in the phagocytic cup. Representative plots are shown in Fig S13. Bar, 10 μm. The graph shows the ratio (mean ± SEM) of the fluorescence peaks at the cup to those on the opposite side of the cell, from n = 8–12 cells. **(B)** F-actin accumulation at the cup was measured for the indicated mutants using fluorescent phalloidin and 18–25 cells. One-way ANOVA compared with WT (*$P < 0.05$; **$P < 0.01$; ***$P < 0.001$; ****$P < 0.0001$).
Source data are available for this figure.

F-actin polymerization. Even in actively phagocyting macrophages, Syk is largely a cytosolic protein (Fig 10) and only a minor fraction of Syk (~10% of cellular Syk) is recruited to the phagocytic cup (Fig 10A) and can be palmitoylated.

Sulfenylation is another, albeit less studied, posttranslational modification of Cys residues (Wages et al, 2015). Our results suggest that mSyk-Cys602 is sulfenylated, but, unlike palmitoylation, we observed that Syk is desulfenylated upon phagocytosis.

**Table 1.  Summary of the mSyk-mutant phenotypes.**

| mSyk mutant | C591S | C602S | RK/AA |
|---|---|---|---|
| Palmitoylation | – | + | + |
| Sulfenylation/desulfenylation | – | – | +/– |
| mSyk phosphorylation on Tyr519/520 | – | – | – |
| Catalytic activity in cellulo | + | – | + |
| mSyk recruitment at the phagocytic cup | – | + | + |
| DHHC5 recruitment at the phagocytic cup | – | – | + |
| Rac1 recruitment at the phagocytic cup | + | – | + |
| Cdc42 recruitment at the phagocytic cup | – | – | + |
| PIP3 and DAG production at the cup | – | – | + |
| Actin polymerization at the phagocytic cup | – | – | +/– |
| FcγR-mediated phagocytosis | +/– | – | +/– |

The C591S mutant is non-palmitoylable, the C602S mutant is non-sulfenylable, and the RK/AA mutant (R219A/K221A) is unable to bind PIP3. Residue numbering refers to mSyk.

Desulfenylation might take place during the drop in $H_2O_2$ production that occurs after ~3 min of phagocytosis (Fig 7). Because the non-palmitoylable mutant (mSyk-C591S) is not desulfenylated, Syk desulfenylation likely takes place after palmitoylation, when Syk is already stably associated with the phagocytic cup. Molecular dynamics studies indicated that Syk desulfenylation facilitates molecular motions and, in particular, enhances the mobility and the exposure of a loop formed by residues 267–325 and containing the first Syk-Tyr that becomes phosphorylated upon activation (Tyr296) (Mansueto et al, 2019).

Syk sulfenylation takes place on mSyk-Cys602 (hSyk-Cys608) within a redox motif that was observed to regulate Zap70 activity (Thurm et al, 2017). For both Syk and Zap70, mutation of the sulfenylated Cys resulted in decreased expression and catalytic activity. Because mutation of the sulfenylated Syk-Cys does not prevent Syk palmitoylation (Fig 2B), the redox state of Syk does not affect palmitoylation, unlike CD81 for instance (Clark et al, 2004). Syk redox sensor is most likely responsible for the activation of Syk by $H_2O_2$, as observed in B cells (Schieven et al, 1993). Because the conformation and activity of Src kinases were also found to be regulated by sulfenylation (Heppner et al, 2018), sulfenylation/desulfenylation is likely a key regulator of phagocytosis efficiency. It is not clear whether sulfenylation of these effectors is concomitant with $H_2O_2$ production at the phagocytic cup.

Membrane association of Syk at the phagocytic cup and, most presumably, on ITAM-bearing receptors, in general, is thus a complex process involving at least three different interactions. The initial recruitment on p-ITAMs is performed by Syk-SH2 domains, allowing palmitoylation. Because the mutant unable to bind PIP3 is palmitoylated, PIP3 binding likely takes place after palmitoylation. Syk recruitment at the cup enables PIP3 production, indicating that an amplification loop may exist, favoring Syk-PIP3 binding and further PIP3 production. Syk recruitment at the cup therefore favors two mechanisms stabilizing its membrane interaction, PIP3 production, and DHHC5 accumulation, which will ensure Syk palmitoylation.

Rac1 and Cdc42 regulate F-actin polymerization at the phagocytic cup, but their role in pseudopod extension is different (Massol et al, 1998; Hoppe & Swanson, 2004). The molecular basis for this differential activation pattern is not clear, although it is probably linked to their recruitment by different phosphoinositides, that is, PIP3 for Rac1 and PI(4,5)P2 for Cdc42 (Heo et al, 2006). We observed here that Syk differentially regulates the recruitment of these GTPases. Results on the non-palmitoylable Syk indicated that Rac1 recruitment at the phagocytic cup does not require Syk palmitoylation, whereas Cdc42 recruitment does require the presence of palmitoylated Syk (Fig 11). These results indicate that Syk could be involved in the sequential recruitment and activation of these GTPases at the phagocytic cup.

In conclusion, we suggested here that Syk undergoes two novel posttranslational modifications that take place on its two last Cys residues. The penultimate Cys is palmitoylated, and the last Cys is likely sulfenylated/desulfenylated. These modifications strongly affected Syk activation and activity during phagocytosis.

# Materials and Methods

### Antibodies, plasmids, oligos, and chemicals

The reagent list is presented in Table S1.

### Plasmids and Syk expression levels

The mSyk expression vector was obtained from Addgene (#50045). FLAG-mSyk was prepared by inserting a FLAG epitope (DYKDDDDK) upstream of Syk coding sequence in the pCi vector (Promega). To prepare EGFP-mSyk, mSyk was inserted downstream of EGFP in pEGFP-C1 (Clontech). Mutations were introduced using QuikChange Lightning Site-Directed Mutagenesis Kit (# 210518; Agilent), and coding sequences were entirely sequenced to check mutations. Syk C591S and Syk C602S were poorly expressed compared with WT (Fig S2), and we adapted our protocols to compare the activity of these mutants with that of WT Syk. Western blots were exposed longer for mutants, except after immunoprecipitation that enabled to obtain comparable amounts of WT and mutant Syk forms. For biochemistry experiments, we used the ratio (+phagocytosis)/(–phagocytosis) to express results so that that the reading was independent of the total amount of the Syk protein. For microscopy experiments, we compared cells expressing similar levels of Syk (see Figs 10A and S1 for instance).

### Macrophages

RAW 264.7 mouse macrophages were obtained and cultured according to the recommendations of the American Tissue Culture Collection. Cells were checked monthly for the absence of mycobacterial contamination. They were transfected with plasmids using Jet-Optimus (Ozyme) as recommended by the manufacturer, and harvested 24 h after transfection. For siRNA transfection, the RNAiMAX reagent (Thermo Fisher Scientific) was used as recommended by the manufacturer. A mix of four different siRNAs were

used against DHHC5 and DHHC20. Cells were harvested 24 h after transfection.

For the preparation of human monocytes, freshly drawn human blood was obtained from the local blood bank (Etablissement Français du Sang, Montpellier, agreement # 21PLER2019-0106). Peripheral blood mononuclear cells were prepared by density gradient separation on Ficoll–Hypaque (Eurobio) before isolating monocytes using CD14 MicroBeads (Miltenyi Biotec). Monocytes were then matured to macrophages by cultivation for 6–8 d in RPMI supplemented with 10% FCS and 50 ng/ml of macrophage colony-stimulating factor (ImmunoTools).

To prepare ΔSyk cells, we used the Addgene all-in-one CRISPR/Cas9 vector (#79144) as described previously (Giuliano et al, 2019). Briefly, three sgRNAs were designed to target *mSyk* using CRISPick. Control vectors targeting *firefly* luciferase were obtained from Addgene (# 80248/80173/80261). Oligos were annealed and phosphorylated before ligation into the plasmid and sequencing. The CRISPR plasmids were then transfected into RAW 264.7 cells. After 24 h, cells were sorted using EGFP fluorescence and an ARIA IIu (Becton Dickinson) cell sorter into 96-well plates (1 cell/well). Clones grew in 2–4 wk. They were amplified and checked for Syk expression using Western blots. We respectively obtained five and seven clones for ΔSyk and ΔLuc cells, and one of each was used for experiments.

## Phagocytosis assays

To assay FcγR-mediated phagocytosis, latex beads (LB30, 3 $\mu$m; Sigma-Aldrich) were opsonized ($10^6$ beads/$\mu$l) using 1 mg/ml mouse IgG (AS10 912; Agrisera) for 45 min at 37°C in PBS. Beads were then washed with PBS and resuspended in PBS/BSA (1 mg/ml).

SRBCs were from Eurobio (SB 068). They were kept in Alsever's solution and used within 3 wk. For opsonization, $1 \times 10^6$ SRBCs/$\mu$l in opsonization buffer (50% DMEM/50% PBS/1 mg BSA/ml) received 1 mg/ml of rabbit anti-SRBCs (MP Biomedicals). After 1 h at 37°C, opsonized SRBCs were washed and finally resuspended in opsonization buffer.

To assay phagocytosis, macrophages were plated on coverslips, washed twice with prewarmed DMEM containing 10 mM Hepes, then overlaid with opsonized latex beads (50/macrophage) or SRBCs (10/macrophage). Phagocytosis was synchronized by centrifugation for 1 min at 300*g* before incubating the plates at 37°C for 5 min (SRBCs) or 15 min (for latex beads). Plates were then placed on ice and washed before labeling extracellular beads using Alexa Fluor 647 donkey anti-mouse IgG (Jackson ImmunoResearch) for 20 min at 4°C (Debaisieux et al, 2015). Cells were then washed, fixed with 3.7% PFA, and mounted using Vectashield Plus (Eurobio). An upright fluorescent microscope (Zeiss Axio Imager Z2 with a 63x NA 1.4 objective) was used for counting beads. Images were randomly acquired. At least 100 cells were examined for each condition. The mean number of phagocytosed beads was calculated, and results are expressed as a percentage of control cells (mean ± SEM).

To monitor recruitments at the phagocytic cup, cells were transfected with EGFP-DHHC or EGFP-Syk or cotransfected with FLAG-Syk and the indicated fluorescent protein (EGFP-Rac1, EGFP-Cdc42, mCherry-Akt, or EGFP-PKD-C1ab). After phagocytosis of opsonized SRBCs for 5 min, cells were fixed with 3.7% PFA,

permeabilized with 0.2% saponin, then stained with phalloidin-TRITC before mounting and imaging using a Zeiss LSM 880 confocal microscope (with a 63× NA 1.4 objective). Line plots (width: 10 pixels) across the cup and the opposite area of the plasma membrane were generated using Fiji. Enrichment was calculated by dividing the fluorescence peak area at the cup by the fluorescence peak area on the opposite side of the cell. At least six phagocyting cells were used to prepare histograms.

To monitor $H_2O_2$ production at the phagocytic cup, RAW 264.7 macrophages (WT or ΔSyk) in glass-bottom 35-mm petri dishes (ibidi) were transfected with pLifeAct-HyPer7. After 18 h, they were placed in RPMI without phenol red and buffered with 10 mM Hepes, on the stage of an LSM 880 microscope (with a 63× NA 1.4 objective) at 37°C. Opsonized SRBCs were added, and images were recorded for 10 min every 20 s at 530 nm after 420- or 490-nm excitation (Pak et al, 2020). A threshold was applied to eliminate weak fluorescence intensities, before calculating the ratio of intensities (490 nm)/(420 nm) using the image calculator of Fiji.

## Palmitoylation assays

These experiments were performed essentially as described earlier (Wan et al, 2007). Monocyte-derived macrophages or RAW 264.7 macrophages were allowed to phagocytose SRBCs for 5 min at 37°C as described above. Control cells did not receive SRBCs. Cells were harvested using a cell scraper, then lysed on ice with lysis buffer (150 mM NaCl, 1 mM EDTA, 25 mM Hepes, pH 7.2) supplemented with Complete antiprotease (Roche), PhosSTOP (Roche), 1% Triton X-100, 0.5% CHAPS, and 80 nM N-ethylmaleimide (NEM). Lysates were briefly sonicated (3 × 1 sec), proteins were precipitated with CHCl3/MeOH, and the pellet was resuspended in 4SB (150 mM NaCl, 5 mM EDTA, 50 mM Tris, pH 7.4, supplemented with 4% SDS) containing 50 mM NEM. After 2.5 h at RT on a rotating wheel, samples were subjected to three sequential CHCl3/MeOH precipitations to remove NEM. They were split into two. One half was resuspended in buffer H (0.9 M hydroxylamine, pH 7.4, 0.2% Triton X-100, 0.6 mM HPDP-biotin) and the other half in buffer C (50 mM Tris, pH 7.4, 0.2% Triton X-100, 0.6 mM HPDP-biotin). After 1 h on the wheel at RT and two sequential CHCl3/MeOH precipitations, samples were resuspended in 150 mM NaCl, 1 mM EDTA, 25 mM Hepes, pH 7.2, containing 0.2% Triton X-100 and 0.05% SDS, then received streptavidin-agarose (Thermo Fisher Scientific).

Beads were washed and finally resuspended in 30 $\mu$l of reducing SDS–PAGE sample buffer; proteins were separated on 10% acrylamide gels before Western blot. Syk was stained using rabbit polyclonal anti-Syk (CST# 2712) followed by mouse anti-rabbit light chain-peroxidase. Syk phosphorylation on mSyk-Tyr519/520 was followed using CST# 2710 and mouse anti-rabbit light chain-peroxidase.

## Sulfenylation assays

After SRBC phagocytosis or NAC treatment (5 mM NAC for 4 h) as indicated, macrophages were lysed in 1 mM EDTA, 150 mM NaCl, 50 mM Tris, pH 7.5, supplemented with Complete antiprotease (Roche), PhosSTOP (Roche), and 0.5% Triton X-100. Lysates were treated with 1 mM DCP-Bio1 (Hourihan et al, 2016) for 1 h at 4°C on a

rotating wheel, before anti-Syk immunoprecipitation as described below. The biotin moiety of DCP-Bio1 was stained with ExtrAvidin-peroxidase. Syk was stained using rabbit polyclonal anti-Syk (CST# 2712) followed by mouse anti-rabbit light chain-peroxidase.

## Syk immunoprecipitation

Cells were lysed with lysis buffer supplemented with Complete antiprotease (Roche), PhosSTOP (Roche), and 1% Triton X-100. mSyk and hSyk were immunoprecipitated using a rabbit monoclonal anti-Syk (CST# 13198) and a mouse monoclonal anti-hSyk (4D10), respectively. Lysates were incubated with the antibodies for 1 h at 4°C on the wheel, before adding magnetic beads covered by protein A/G (88802; Pierce), and for 1 h at 4°C on the wheel, before washes with lysis buffer, SDS–PAGE on 10% acrylamide gels, and Western blotting.

## Protein preparation for proteomic analyses

ΔSyk RAW 264.7 macrophages were transfected with FLAG-Syk. After 18 h, cells were lysed as described above, before immunoprecipitation using anti-FLAG magnetic resin and SDS–PAGE. Gels were stained using blue silver (Candiano et al, 2004) before excision of the 70-kD band. The gel pieces were successively washed with 25 mM $NH_4HCO_3$ and acetonitrile, reduced using 10 mM DTT in 25 mM $NH_4HCO_3$ (1 h at 57°C), alkylated by 55 mM iodoacetamide in 25 mM $NH_4HCO_3$, and treated with trypsin (12.5 ng/µl) in 25 mM $NH_4HCO_3$, overnight at 37°C. The digested peptides were extracted in 34.9% $H_2O$, 65% acetonitrile, and 0.1% HCOOH, the acetonitrile was removed by evaporation, and peptides were analyzed by nanoLC-MS/MS.

## LC-MS/MS experiments

The analysis was performed using a nanoACQUITY Ultra-Performance LC (Waters). The samples were trapped on a Symmetry C18 pre-column (Waters), and the peptides were separated on an ACQUITY UPLC BEH130 C18 separation column (Waters). The solvent system consisted of 0.1% formic acid in water (solvent A) and 0.1% formic acid in acetonitrile (solvent B). Peptides were applied in solvent A, and elution was performed at 60°C using a gradient from 8% to 40% of B. The mass spectrometer was operated using a spray voltage of 1,800 V and a capillary temperature of 250°C. The MS scan had a resolution of 70,000, the AGC target was 3 × $10^6$, and the maximum IT was 50 ms on m/z [300–1,800] range. The MS/MS scans had a resolution of 17,500, the AGC target was 1 × $10^5$, and the maximum IT was 100 ms with fixed first mass of 100 m/z and isolation window of 2 m/z. Top 10 HCD was selected with MS2 identity threshold of 2 × $10^5$ and dynamic exclusion of 60 s. The normalized collision energy (NCE) was fixed at 27 V. The system was controlled by Thermo Fisher Scientific Xcalibur software (4.0.27.19). Raw data collected were processed and converted with MSConvert into .mgf peak list format. The peak list was searched against a *Mus musculus* combined target–decoy database using Mascot (Matrix Science). Error tolerance on MS was set at 15 ppm for peptides and at 30 ppm for MS/MS fragments. A maximum of one trypsin miscleavage was allowed. Cysteine

carbamidomethylation (58.004 Da) was set as a fixed modification, whereas methionine oxidation (15.995 Da) and cysteine dioxidation (31.972 Da) were set as variable modifications. Proline pipeline (http://www.profiproteomics.fr/proline/) was used to validate the identification results. This statistical validation was performed using the target–decoy approach, which consists in creating a "decoy" sequence that does not exist in nature. The false identifications thus distribute evenly between the real database (target) and the decoy one, and so the amount of decoy hits can be used to estimate the false discovery rate (FDR = decoy hits/decoy hits + target hits) and eliminate false positives. Peptide identification validation parameters were set as follows: a minimal length of seven amino acid, a pretty rank ≤1, and protein validation parameter FDR ≤ 1.

## Assays for Syk catalytic activity

The FRET assay was performed essentially as described previously (Xiang et al, 2011). ΔSyk RAW 264.7 cells on coverslips were transfected with the Syk biosensor and the indicated version of FLAG-Syk. After 24 h, they were allowed to phagocytose IgG-opsonized SRBCs for 5 min as indicated, then washed, and fixed overnight with PFA (3.7%) that was then neutralized with 50 mM ammonium chloride. After washes with PBS, coverslips were mounted in Vectashield plus, then observed using a Zeiss LSM 880 confocal microscope fitted with a 63× NA 1.4 objective. ECFP and FRET fluorescence were recorded at 475 and 530 nm upon excitation at 435 nm, whereas YPet fluorescence was monitored at 530 nm after excitation at 515 nm. The sensor phosphorylation inhibits the FRET between ECFP and YPet with a concomitant increase in ECFP fluorescence (Xiang et al, 2011). Results are expressed as the mean ± SEM (n = 30–35 cells) of the ratio of ECFP/FRET fluorescence.

## RT–qPCR

RT–qPCR was performed as described earlier (Schatz et al, 2023). Briefly, cell RNA was first extracted using TRIzol, then transcribed into DNA using All-in-One RT Master Mix. DHHC20 primers for qPCR were designed using primer blast. qPCR was performed using SYBR Green Master Mix as recommended by the manufacturer. Large Ribosomal Subunit Protein EL27 (RPL27) transcripts were used to normalize data. Primers and reagents are listed in Table S1.

## Molecular dynamics simulations

The crystal structure of Syk (PDB 4FL3) was used to perform molecular dynamics simulations. The missing residues and the sulfenylation of the Cys608 were modeled using Charmm-GUI (Lee et al, 2016). Ligands (AMP-PNP, $Mg^{2+}$, and crystal water molecules) present in the PDB structure were preserved for the simulation. Systems (WT hSyk, hSyk-C608-OH, and Syk-C608S) were immersed in a water box (TIP3P model) surrounding the protein with 20 Å of side edge at a physiological salt concentration ([NaCl] = 0.154 M). Each system was parameterized using the CHARMM36m force field (Huang et al, 2017), and electrostatic neutrality was achieved by completing with few additional Na+ or Cl− ions depending on the

system total charge. Periodic boundary conditions in conjunction with the particle mesh Ewald method were set up for each system. After energy minimization (50,000 steps of conjugate gradients) at 0 K and a gradual heating to 310 K, an equilibration phase of 250 ps was achieved. Production runs (used for the analysis) were carried out in the isobaric–isothermal ensemble, at constant temperature (310 K) and pressure (1 atm), using Langevin dynamics and Langevin piston as implemented in NAMD 3.0α12 (Phillips et al, 2020) for a total period of 500 ns.

Principal component analysis (PCA) based on 3D coordinates from the simulations was achieved using a homemade Python script based on the ProDY package (v2.2.0), allowing for Essential Dynamics Analysis (EDA) (Zhang et al, 2021), which was applied to all frames of the production run for both systems using all CA atoms after superposition of coordinates to the first frame. Root-mean-square fluctuations (RMSF) were deduced from the first three normal modes computed by EDA allowing us to evaluate the global mobility of the residues during the simulation. Trajectory analyses and EDA visualization were performed with the Visual Molecular Dynamics program (VMD, v1.9.4 [Humphrey et al, 1996]).

## Data Availability

All data are available at https://doi.org/10.6084/m9.figshare.30999478.

## Supplementary Information

## Acknowledgements

This work was funded by the CNRS and the French Proteomic Infrastructure (ProFI) project (grant ANR-10-INBS-08 and ANR-24-INBS-0015) to J-M Strub. We are grateful to Stéphane Bodin, Laurence Abrami, and members of the APIR team for discussions and suggestions.

## Author Contributions

M Jansen: investigation.
J-M Strub: data curation, formal analysis, investigation, visualization, and methodology.
L Chaloin: conceptualization, data curation, formal analysis, validation, investigation, visualization, methodology, and writing—review and editing.
P Coopman: conceptualization, validation, visualization, and writing—review and editing.
B Beaumelle: conceptualization, data curation, formal analysis, supervision, validation, investigation, visualization, methodology, project administration, and writing—original draft, review, and editing.

## Conflict of Interest Statement

The authors declare that they have no conflict of interest.

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
