## [Reviewer comments · Life Science Alliance]

Syk activation during FcγR-mediated phagocytosis involves Syk palmitoylation and desulfenylation

Maxime Jansen, Jean-Marc Strub, Laurent Chaloin, Peter Coopman, and Bruno Beaumelle

DOI: <https://doi.org/10.26508/lsa.202503500>

Corresponding author(s): Bruno Beaumelle, Institut de Recherche en Infectiologie de Montpellier

Review Timeline:

Submission Date:	2025-08-29
Editorial Decision:	2025-09-02
Revision Received:	2025-11-14
Editorial Decision:	2025-12-24
Revision Received:	2026-01-09
Accepted:	2026-01-13

Scientific Editor: Tim Fessenden

Transaction Report:

Please note that the manuscript was previously reviewed at another journal and the reports were taken into account in the decision-making process at *Life Science Alliance*.

Referee #1

Report for Author:

Several Tyr kinases, including Src family kinases and ZAP-70, are regulated by Cys sulfenylation creating Cys-SOH, at specific sites. Here, the authors set out to examine the roles of Cys residue modification in regulating the function of the Syk tyrosine kinase in phagocytosis, and in particular determine whether this process is regulated by sulfenylation. By expressing WT, catalytically dead, tandem SH2 binding defective, PIP3-binding site RK/AA mutant, and seven Cys to Ser mutant mouse Syk proteins in Syk knockout mouse RAW246.7 myeloid cells, which are phagocytosis defective, they determined that Ser mutation of C591 or C602, which correspond to the final two Cys in the C-terminal lobe of the Syk catalytic domain, had strongly reduced ability to restore phagocytic activity in Syk KO cells. By using CSS-palm software, they predicted that C591 could be a palmitoylation site, and using an acyl-biotin exchange assay they showed that WT Syk palmitoylation was increased when RAW246.7 cell phagocytosis was induced, and that the palmitoylation of WT but not C591S was increased in Syk-re-expressing KO cells. Several DHCC family palmitoylating enzymes are known, and by using siRNA depletion of DHCCs, they found that DDHC5 is primarily responsible for Syk palmitoylation., with DDHC20 possibly playing some upstream role. Quantification of the defects in phagocytosis efficiency revealed that only DHCC5 knockdown reduced phagocytosis, and they showed that Syk and DDHC5 co-precipitated during phagocytosis. While these findings established a role for C591 palmitoylation in phagocytosis, it left the question of what role C602 plays in phagocytosis unanswered. This led the authors to test if C602 is sulfenylated. Using hMDM cells and the DCP-Bio1 specific reagent for biotinylation of sulfenylated Cys-SOH, they showed that Syk immunoprecipitated from DCP-Bio1-treated human hMDM cell extracts was labeled with biotin. The level of Syk sulfenylation in these cells was decreased ~20% when phagocytosis was induced with opsonized SRBCs, and the same effect was observed in RAW246.7 cells. Further analysis showed that the RK/AA, C591S and C602S mutant Syk's were not desulfenylated upon induction of phagocytosis, from which they concluded that palmitoylation, sulfenylation and PIP3 binding are all required for phagocytosis. MDS simulations of WT and C602-SOH Syk catalytic domain revealed that desulfenylation of C602-SOH would increase the mobility of a flexible loop comprised of residues 267-325, allowing access for Src to phosphorylate Y348/Y352 upstream of the catalytic domain. Using the series of Syk mutants they went on to show that SH2 binding, desulfenylation, palmitoylation and PIP3 binding are all important for Syk catalytic activity, based on the level of activating autophosphorylation of Y525/Y526 in the activation loop. Also, using a FRET-based live cell Syk activity assay they showed that C602S Syk was inactive, whereas C591S and RK/AA mutant Syk's were catalytically active. Palmitoylation of S591 was also shown to be important for Syk localization to the phagocytic cup, whereas PIP3 binding and sulfenylation were not. Furthermore, Syk sulfenylation/desulfenylation and palmitoylation were required for PIP3 and DAG production at the phagocytic cup. Finally, they showed that both palmitoylation and sulfenylation, as well as PIP3 binding and kinase activity, were required for Rac1 and Cdc42 recruitment to the cup, as well as concomitant actin polymerization at the phagocytic cup. Based on these new findings about Cys posttranslational modification of Syk's phagocytic functions, the authors developed a revised model of the role of Syk in this process.

These new findings showing that palmitoylation of C591 and sulfenylation/desulfenylation of C602 in the Syk tyrosine kinase catalytic domain are required for its function in macrophage phagocytosis downstream of FcγR activation are intriguing, and is potentially a useful advance in our understanding of how Syk is activated through the FcγR to drive phagocytosis. However, there are weaknesses: some of the data supporting their conclusions are not totally compelling with significant variability in Syk protein levels within the same experiment weakening their quantitative analysis. There is a lack of mechanistic insight, particularly with regard to sulfenylation and desulfenylation. For instance, how is the apparently constitutive sulfenylation of C602 mediated, and what is the stoichiometry of this modification (is the level of C602 sulfenylation increased when cells are exposed to an oxidizing agent - apparently Syk can be activated by H₂O₂ treatment of B cells, but this would be expected to increase sulfenylation). Likewise, how is C602 desulfenylation carried out; is C602 sulfenylation reversed by treating cells with an anti-oxidant such as NAc-Cys (what enzymes are involved?), and is desulfenylation selective for Syk, or are other proteins in the cell desulfenylated upon activation of FcγR? Another weakness is

that the relative stoichiometries of the C591 and C602 modifications have not been determined, nor is it clear that whether the two modifications are present simultaneously on a single Syk molecule (stoichiometries could be measured by carrying out streptavidin pulldown after biotin exchange for both modifications and then blotting both the streptavidin-captured proteins and the supernatants for Syk followed by quantitation of the ratios). The authors state that Syk palmitoylation stoichiometry is low, but did not directly measure it. With regard to the stoichiometry of Syk sulfenylation, they showed a 20% reduction in the level of sulfenylation upon FcγR activation, but it is not clear what fraction of the total Syk population is sulfenylated at C602 to start with. They imply that desulfenylation might be restricted to palmitoylated Syk molecules (possibly Palm-Syk molecules recruited to P-ITAMs), but do not provide any evidence for this or how the desulfenylation event might be localized to the phagocytic cup. In this regard, they do not provide direct biochemical evidence that C602 is sulfenylated, but rather use the fact that the C602S mutant abolishes sulfenylation as measured with DCP-Bio1 to conclude that it is C602 that is modified. Even though C602 lies in a "redox motif", it would be reassuring to carry out peptide proteomic MS analysis on the isolated DCP-Bio1 biotinylated Syk protein to show that biotin is coupled to the C602 residue. With regard to C591 palmitoylation, how is DDHC5-driven palmitoylation of C591 stimulated upon ligand activation of FcγR. For instance, how does FcγR signaling induce DDHC5 association with Syk once it interacts with the phospho-ITAMs on FcγR, and is the observed interaction of Syk with DDHC5 direct? Also, is the palmitate on C591 turned over or is it stable?

Points: 1. Figure 2B: Given the low levels of C591S and C602S protein expression, this experiment needs a control showing the levels of Syk protein in these samples. In panel B the level of palmitoylated C591S Syk was as high as in the WT Syk sample, which is odd if the level of the C602S protein was much lower. The reason for the strong "palmitoylation" signals observed with the C591S is not clear - is the mutant protein S-acylated at some other site? The authors conclude from these data that Syk is palmitoylated at C591, but to me the logic is flawed.

2. Figure S2: The low level of C591S and C602S protein expression was noted, but the reason for their low levels was not explained. Are they turned over rapidly and stabilized by a proteasome inhibitor? If so, does this mean that they are partially unfolded?

3. Figure 3A-D: The data in these panels are hard to interpret. DHCC20 knockdown increased the level of DHHC5 only in SRBC-activated cells, but we are not shown a blot for how much the DHHC20 levels were reduced by the DHHC20 siRNA treatment. Moreover, essential controls with the re-expression of siRNA resistant forms of DHHC5 and DHCC20 to show that the observed siRNA effects were on target are missing. In panel C, DHHC5 knockdown apparently reduced SRBC-induced palmitoylation of Syk, although Syk protein level was also reduced. Moreover, there was no increase in Syk palmitoylation in the Luc control pair, with SRBC treatment. The reduction in phagocytosis with DHHC5 knockdown was barely significant, and this experiment lacks an siRNA-resistant re-expression DHHC5 control.

4. Figure 3E: The inhibitory effect of massive overexpression EGFP-mDHHC20 is quite striking and a better explanation is needed for why overexpression of decreased Syk palmitoylation by 3-fold. They say "This is probably because EGFP-mDHHC20 overexpression decreased endogenous DHHC5 level by ~60 %, a drop that could be due to over-palmitoylation of mDHHC5". To be sure about this, they would need to measure the palmitoylation stoichiometry. Is DHCC5 RNA level reduced by 60% with overexpression of DHHC20? Finally, the apparent decrease in Syk palmitoylation could be explained by the significantly lower level of Syk protein in the unstimulated cells compared to that in the SRBC-treated cells; how could the level of Syk protein increase so rapidly upon treatment?

5. Figure 4A: Why were so many sulfenylated bands observed in the hSyk IP sample?

6. Figure 4B: The data identifying C602 as a sulfenylation site in RAW246.7 cells are not very convincing due to the fuzziness of the biotin positive bands, which have a very high smeary background, which will make it hard to accurately quantify the embedded band of interest, and therefore it is hard to be confident of the sulfenylation values that are recorded in the histogram. Moreover, the intensities of the biotinylated WT and C602S bands look similar, and, if one takes into account that the level of the C602S mutant Syk protein was significantly lower

than that of the WT Syk protein, the actual degree of sulfenylation would seem to be higher for C602S mutant Syk! In other words, based on these data the evidence that C602 is selectively sulfenylated does not seem to be very strong. Also, the level of the RK/AA Syk protein was extremely low especially in the unstimulated cell sample, whereas the biotinylated band was increased in this sample. Based on the quantitation below the blots, these samples were done in quadruplicate, but the authors either need to repeat this or show more convincing blots where the level of Syk protein are more consistent. In this regard, the authors need to indicate in the legends of all the figures with samples of this sort whether these are technical or biological repeats, and how many times these experiments were repeated. If the current results are correct, then one also needs an explanation for why explain why the RK/AA and C591S mutants were not desulfenylated - does this process only occur when the Syk protein is recruited to the phagocytic cup, and if so what is the mechanism?

7. Figure 5: Given the relatively limited amount of information on the MDS analysis and the mechanism through which desulfenylation leads to catalytic activation, it would certainly strengthen the authors case if they had some experimental HDMX data to show that desulfenylation affects the exposure of residues 267-325 as they propose. Did the authors also carry out MDS on the S602 mutant Syk catalytic domain. What about using AlphaFold 3 for static modeling? The two 3D structure panels shown in panel B need better labeling to describe which regions of the catalytic domain we are looking at, including the Y348/Y352 residues, if they are in the model. Also, the ball and stick shapes depicting C602 and C602S-OH need to be labeled and described in the legend, and possibly an enlargement of this region shown as an inset. What specific interactions does the sulfenyl oxygen make in the model?

8: Figure 6: The activity data look reasonable, but again the large variability in the levels of Syk protein within control/stimulated pairs, makes it hard to be totally convinced by the quantitation.

9. Figure 7: Where is the FRET biosensor protein localized in the cells, and could this affect the interpretation of this experiment? Also, there were SRBC-stimulation induced increases in the activities of the RK/AA and C591S mutant Syk's, whereas in Figure 6 the activities of the RK/AA and C591S mutants were as low as those for the SH2 binding and C602S Syk mutants, which here showed no activation. Is there an explanation for these discrepancies?

10. Figure 8: Why was the kinase-dead Syk mutant not recruited to the cup - is this because it does not activate PIP3 production? Sulfenylation and palmitoylation data for this mutant are not shown.

11. Figure 10A/B: There is a huge scatter range for the WT datasets, and this means that the plots for the mutants are very compressed. Is there a better way of depicting these data?

12. Figure 11: Their new model is not very informative - it does not depict any conformational changes in the Syk catalytic domain upon desulfenylation or activation loop phosphorylation, and in this regard the whole Syk protein is shown as a blob without a discrete Syk catalytic domain and tandem SH2 domains, with the tandem P-ITAM phosphates that engage the tandem SH2 domains shown at either end of the Syk blob. Also, the associated "Src" molecule apparently has two palmitoyl groups attached, when the only c-Src lipid modification is a single N-terminal myristoyl group, with some of the other SFKs having an N-terminal myristoyl group and a palmitoylated Cys within the first 10 residues. Also, the model implies an order of Syk modification and binding. that was not fully demonstrated experimentally. Is sulfenylated Syk selectively recruited to the FcγR as implied in step 1? The model lacks the Src-mediated phosphorylation of Y348/Y352 step, whose phosphorylation initiates activation.

13. For other tyrosine kinases, it is known that an intramolecular S-S bond can form between Cys residues in the catalytic domain. Did the authors consider if C591 and C602 could form S-S bonds - C591 and C602 are too far apart in the Syk catalytic domain structure to form an S-S bond, but C602 might be able to form an S-S bond with another Cys, such as Cys587, which might be possible based on the crystal structure.

Report for Author:

In the manuscript by Jansen et al., the authors address an important question about how multiple modifications might regulate Syk function, and they present some intriguing findings. However, several technical and interpretative issues need to be addressed before the work is suitable for publication.

My primary concern relates to the palmitoylation detection method. In Figure 2A, I noticed that the negative control lane (without hydroxylamine) shows comparable or even stronger signal than the experimental condition. Since the ABE assay fundamentally depends on hydroxylamine to cleave thioester bonds before biotinylation, this suggests significant non-specific binding or technical issues with the assay. This needs to be resolved, as it affects interpretation of all the palmitoylation data. Additionally, this key figure lacks quantification - it would strengthen the conclusions to show replicate experiments with statistical analysis demonstrating phagocytosis-induced palmitoylation.

A similar issue with quantification appears in several other figures. For example, Figure 3F shows a co-immunoprecipitation between DHHC5 and Syk, but without quantification across multiple experiments, it's difficult to assess the significance of this interaction. Given the importance of these biochemical findings to the overall story, providing quantitative data from replicate experiments would substantially strengthen the manuscript.

I'm also concerned about the biological timing of the sulfenylation measurements. The authors report Syk desulfenylation at 5 minutes post-phagocytosis, but this precedes the major oxidative burst, which typically peaks around 30-60 minutes when NADPH oxidase is fully active. This raises important questions: Is the observed desulfenylation related to loss of basal modification rather than oxidative burst-dependent changes? It would be valuable to extend the time course to capture the actual oxidative burst period and include NADPH oxidase inhibitors to test ROS dependence. This would help clarify the biological context of the modification. The manuscript would also benefit from more precise language regarding the necessity of these modifications. While the authors state that palmitoylation is "strictly necessary" for phagocytosis, Figure 1B shows that the C591S mutant retains approximately 30% activity. This suggests the modification is important for optimal function rather than absolutely required. Adjusting the language throughout to reflect these partial phenotypes would provide a more accurate representation of the data.

An important technical consideration is the reduced expression of the C591S and C602S mutants shown in Figure S2. Since these mutants express at substantially lower levels than wild-type, this could contribute to their functional deficits. It would be helpful to either normalize functional data to expression levels or discuss how stability effects might be part of the regulatory mechanism for these modifications.

The siRNA experiments in Figure 3 would be strengthened by including rescue controls. While the knockdown data suggests DHHC5 involvement, the partial knockdown and compensatory DHHC20 upregulation complicate interpretation. Rescue with an siRNA-resistant DHHC5 would provide stronger evidence for specificity. The unexpected finding that DHHC20 overexpression decreases palmitoylation (Figure 3E) is interesting and might warrant further investigation into the regulatory relationships between these enzymes.

The molecular dynamics simulations, while providing some structural insights, are limited by the absence of membrane context and the relatively short timescale. Since palmitoylation mediates membrane association, including membrane in the simulations would provide more physiologically relevant information. Alternatively, the limitations of the current simulations should be acknowledged more explicitly.

Additionally, there are inconsistencies between assays (how can C591S have 30% phagocytosis but almost no F-actin accumulation?), the poor quality of the sulfenylation blots, and the oversimplified model in Figure 11 that ignores all the partial phenotypes.

Despite these concerns, the manuscript contains several valuable observations. The discovery that Syk undergoes both palmitoylation and sulfenylation during phagocytosis is novel and important. The differential effects on Rac1 versus Cdc42 recruitment are particularly intriguing and suggest these modifications might fine-tune different aspects of phagocytic signaling, but it's built on a shaky technical foundation.

While the study identifies novel Syk modifications with potential immunological significance, methodological limitations and overinterpretation of partial phenotypes weaken its conclusions. Addressing these concerns through additional validation and mechanistic experiments would solidify the proposed model and enhance its impact on the field of phagocytosis signaling.

Referee #3

Report for Author:

The manuscript by Jansen et al., describes novel mechanisms of activation of the tyrosine kinase SYK that plays an important role downstream of the FcγR-mediated phagocytosis. In particular the study focuses on two post-translational modifications, palmitoylation and sulfenylation. Palmitoylation of SYK is shown to be mediated by the S-acyl transferase DHHC5 and to be necessary for SYK recruitment to the phagocytic cup, activation and downstream events such as Cdc42 activation and F-actin polymerisation. Sulfenylation was proposed to play an inhibitory role as desulfenylation occurs upon FcR-phagocytosis and molecular dynamics experiments suggested that it could facilitate phosphorylation of the kinase.

This is a very interesting manuscript that really extends our knowledge on the modalities of regulation of a kinase that is crucial in many phagocytic processes.

There are a few points that should be clarified by the authors.

General comment: the figures could be re-organised to be more compact and leave less white spaces in order to give a better general overview of the results.

Specific comments on the figures

Figure 1A: the authors use SYK knock-out cells: how many clones were used in the study?

Figure 1B: there seem to be only 2 data points in the graph: how were the statistics generated?

Figure 3E: same question

Figure 3C: the quantification performed is not well described and the legend of the Y axis is also unclear: what does +/- phago mean?

Figure 8: the recruitment of SYK at the phagocytic cup in the WT and RK/AA conditions does not seem to be a very local event, as one can tell from the image presented. Could the authors comment on this? In addition, image analysis and quantification of the recruitment was performed on 6-9 cells only, and it is not mentioned from how many independent experiments. This result should be improved with more data points.

Figure 8, 9, 10: could the authors provide the blue images in cyan or better, in black and white to improve their quality ?

Figure 11: there is a problem of scale for the Ig and the red blood cell. As some experiments were performed with beads, the summary scheme could be represented with only a portion of a larger bead/ particle. More importantly, the Fc portion does not bind to the FcR as it is represented. The representation of the results obtained in the manuscript should be indicated with colors different from the mechanisms that are still hypothetical.

Minor comments:

- RAW cells should be RAW264.7 cells in the text

There are some typos :

- p4 Line 29: we observed using that DHHC5 specifically associates with Syk

- p4 the title: DDHC5: should be replaced by DHHC5

- supp fig 4 to 7 legends: Representative line plots (width: ten pixels) across cells, enabling to monitor XXX compared to a non-phagocytic region

September 2, 2025

Re: Life Science Alliance manuscript #LSA-2025-03500-T

Dr. Bruno Beaumelle
IRIM
UMR 9004 Université de Montpellier-CNRS
1919 route de Mende
MONTPELLIER 34293
France

Dear Dr. Beaumelle,

Thank you for transferring your manuscript entitled "Syk activation during FcγR-mediated phagocytosis involves Syk palmitoylation and desulfenylation" to Life Science Alliance. As we noted in our offer inviting consideration at LSA, we invite you to submit a revised manuscript with the following revisions noted below. We will contact the same reviewers from another Journal to evaluate the revised manuscript.

- Incorporate all quantifications, validation assays and control experiments as stipulated in the draft rebuttal letter.
- Adjust the main claims to acknowledge the observations that remain preliminary, if needed by noting limitations in methodology to detect and perturb sulfenylation. In particular tone down the claims on the significance of desulfenylation at C602.

Please contact me with any questions about these requested changes or the ensuing re-review process.

Thank you for this interesting contribution to Life Science Alliance. We are looking forward to receiving your revised manuscript.

Sincerely,

-- Summary blurb (enter in submission system): A short text summarizing in a single sentence the study (max. 200 characters including spaces). This text is used in conjunction with the titles of papers, hence should be informative and complementary to

the title and running title. It should describe the context and significance of the findings for a general readership; it should be written in the present tense and refer to the work in the third person. Author names should not be mentioned.

B. MANUSCRIPT ORGANIZATION AND FORMATTING:

To Tim Fessenden
Scientific Editor
Life Science Alliance

Dear Dr Fessenden,

Please find attached our revised manuscript entitled " Syk activation during FcγR-mediated phagocytosis involves Syk palmitoylation and desulfenylation " (#LSA-2025-03500-T).

We first would like to thank the reviewers for their positive comments and their constructive suggestions that helped us to strengthen the manuscript.

We revised the manuscript following your recommendations and those of the reviewers. We performed not only the key experiments suggested by the reviewers but also additional ones.

The main new data we show are

- the recruitment of DHHC5 to the phagocytic cup (suggested by reviewer 1)
- the evidence for Syk-Cys602 oxidation obtained by proteomic analysis (suggested by reviewer 1)
- the inhibition of Syk sulfenylation by N-acetyl Cys (suggested by reviewer 1)
- the characterization of the sulfenylated Cys environment (suggested by reviewer 1)
- the wave-like production of H₂O₂ at the phagocytic cup (to answer to reviewer 2).

We carefully addressed each point raised by the reviewers in the point-by-point response below. Referee's comments are in black and our responses are in blue.

Referee #1

There is a lack of mechanistic insight, particularly with regard to sulfenylation and desulfenylation. For instance, how is the apparently constitutive sulfenylation of C602 mediated, and what is the stoichiometry of this modification (is the level of C602 sulfenylation increased when cells are exposed to an oxidizing agent - apparently Syk can be activated by H₂O₂ treatment of B cells, but this would be expected to increase sulfenylation).

There are only a few studies on protein sulfenylation. For instance, a PubMed search returns 338 responses compared to 11,466 for protein palmitoylation. It is difficult to assess the stoichiometry of this modification. Even the most comprehensive studies on protein sulfenylation (Paulsen *et al*, 2011) did not report any sulfenylation stoichiometry. In the case of Syk, which is present in many subcellular compartments, this information would not be very informative.

Regarding the H₂O₂ production we now show that H₂O₂ is produced very early after the onset of phagocytosis (Fig.7, Video1) and H₂O₂ is known to activate Syk (Qin *et al*, 1998). This H₂O₂ production drops 3 min after the start of phagocytosis, and this is kinetically consistent with the Syk desulfenylation we observed.

Likewise, how is C602 desulfenylation carried out; is C602 sulfenylation reversed by treating cells with an anti-oxidant such as NAc-Cys (what enzymes are involved?), and is desulfenylation selective for Syk, or are other proteins in the cell desulfenylated upon activation of FcγR?

We were surprised to observe that many sulfenylated proteins associated with Syk (Fig5A). The NAC treatment proposed by the reviewer was effectively an important control for the specificity of DCP-Bio1 labeling (Krasnowska *et al*, 2008). We now show that pretreating cells with NAC severely inhibited Syk sulfenylation (Fig.5B). This study is focused on the Syk protein.

Another weakness is that the relative stoichiometries of the C591 and C602 modifications have not been determined, nor is it clear that whether the two modifications are present simultaneously on a single Syk molecule (stoichiometries could be measured by carrying out streptavidin pulldown after biotin exchange for both modifications and then blotting both the streptavidin-captured proteins and the supernatants for Syk followed by quantitation of the ratios). The authors state that Syk palmitoylation stoichiometry is low, but did not directly measure it.

We already performed the experiment proposed by the referee, and all our results are normalized to Syk levels. This is particularly important when using mutants that displayed different expression levels, as we experienced. The methods allowing quantification of palmitoylation stoichiometry are the acyl-PEG (Yucel *et al*, 2024) and acyl-RAC (Werno & Chamberlain, 2015) techniques. We tried them but they were not applicable to Syk, probably because of Syk low-palmitoylation efficiency, due to the large pool of cytosolic Syk (cf Fig.10A).

We could not directly measure the efficiency of Syk palmitoylation but, since Syk accumulation at the cup requires palmitoylation, the percentage of cellular Syk that is present at the cup (~10%) provides a maximum value for Syk palmitoylation.

With regard to the stoichiometry of Syk sulfenylation, they showed a 20% reduction in the level of sulfenylation upon FcγR activation, but it is not clear what fraction of the total Syk population is sulfenylated at C602 to start with.

As discussed above, sulfenylation stoichiometry cannot be accurately quantified.

They imply that desulfenylation might be restricted to palmitoylated Syk molecules (possibly Palm-Syk molecules recruited to P-ITAMs), but do not provide any evidence for this or how the desulfenylation event might be localized to the phagocytic cup.

The evidence is indirect. Since Syk desulfenylation requires palmitoylation (the palmitoylation-defective mutant is not sulfenylated) and palmitoylation takes place at the phagocytic cup, desulfenylation most likely takes place at this level. The wave-like H₂O₂ production at the cup is consistent with the sulfenylation / desulfenylation switch we observed for Syk.

In this regard, they do not provide direct biochemical evidence that C602 is sulfenylated, but rather use the fact that the C602S mutant abolishes sulfenylation as measured with DCP-Bio1 to conclude that it is C602 that is modified. Even though C602 lies in a "redox motif", it would be reassuring to carry out peptide proteomic MS analysis on the isolated DCP-Bio1 biotinylated Syk protein to show that biotin is coupled to the C602 residue.

We agree with the referee that it was important to confirm sulfenylation biochemical data. We now show proteomic data indicating that mSyk is oxidized on Cys602 (Fig. S6).

With regard to C591 palmitoylation, how is DDHC5-driven palmitoylation of C591 stimulated upon ligand activation of FcγR. For instance, how does FcγR signaling induce DDHC5 association with Syk once it interacts with the phospho-ITAMs on FcγR, and is the observed interaction of Syk with DDHC5 direct? We know show that DDHC5 is concentrated at the phagocytic cup while DDHC20 is not. This accumulation of DDHC5 at the cup requires the presence of catalytically active and palmitoylable Syk. Regarding the Syk-DDHC5 interaction, we do not know whether it is direct. It might require cellular chaperones as we observed earlier for the HIV Tat-DDHC20 interaction (Chopard *et al*, 2018).

Also, is the palmitate on C591 turned over or is it stable?

The ABE technique only enabled us to detect Syk palmitoylation. As discussed elsewhere (Gao & Hannoush, 2018), contrarily to click chemistry, the ABE technique is not suitable to follow palmitate turn-over on proteins.

Points: 1. Figure 2B: Given the low levels of C591S and C602S protein expression, this experiment needs a control showing the levels of Syk protein in these samples.

We added the inputs as requested.

In panel B the level of palmitoylated C591S Syk was as high as in the WT Syk sample, which is odd if the level of the C602S protein was much lower. The reason for the strong "palmitoylation" signals observed with the C591S is not clear - is the mutant protein S-acylated at some other site? The authors conclude from these data that Syk is palmitoylated at C591, but to me the logic is flawed.

We repeated this experiment and the palmitoylation levels of the C591S Syk, both with and without phagocytosis were much lower.

2. Figure S2: The low level of C591S and C602S protein expression was noted, but the reason for their low levels was not explained. Are they turned over rapidly and stabilized by a proteasome inhibitor? If so, does this mean that they are partially unfolded?

We do not know why these mutants are unstable. Replacing a sulfur by an oxygen usually do not have such major effects on protein levels. We effectively tried a proteasome inhibitor (MG132, now shown in Fig.S2) as well as inhibitors of lysosomal degradation (bafilomycin A, not shown) but they all failed to stabilize the C591S and C602S mutants. Regarding the conformation of these mutants, MD studies showed that mSyk C602S (C608S for hSyk) is more tightly packed than the WT form (Fig. 6; and see below).

3. Figure 3A-D: The data in these panels are hard to interpret. DHHC20 knockdown increased the level of DDHC5 only in SRBC-activated cells, but we are not shown a blot for how much the DDHC20 levels were reduced by the DDHC20 siRNA treatment.

All the anti-DDHC20 antibodies we tried (including those that provide good signals on human cells) failed to detect DDHC20 in RAW cells. This is why we performed qRT-PCR to show that the siRNA DDHC20 decreased DDHC20 expression by ~60% (Fig.3B). The relationship between DDHC5 and DDHC20 is indeed complex and we did our best to clearly explain our data.

Moreover, essential controls with the re-expression of siRNA resistant forms of DDHC5 and DDHC20 to show that the observed siRNA effects were on target are missing.

We now dispose of four types of observations indicating that Syk palmitoylation is performed by DDHC5: the siRNA depletion consequences, the overexpression effect, the co-immunoprecipitation, and the Syk-dependent recruitment of DDHC5, but not DDHC20 at the cup. Moreover, ectopic expression of DDHC20 decreases DDHC5 level and Syk palmitoylation just as siRNAs targeting DDHC5 (Fig.3), thereby confirming the DDHC5 siRNA effect. To our opinion, expression of siRNA-resistant forms of DDHC does

not seem to essential to confirm the siRNA effects that are already confirmed by DHHC overexpression. Please note that we used On-Target Plus siRNAs from Dharmacon, a mixture of 4 siRNAs.

In panel C, DHHC5 knockdown apparently reduced SRBC-induced palmitoylation of Syk, although Syk protein level was also reduced. Moreover, there was no increase in Syk palmitoylation in the Luc control pair, with SRBC treatment.

As shown in Fig.3C, for the Luc siRNA, there was a 40 % increase in Syk palmitoylation induced by phagocytosis.

The reduction in phagocytosis with DHHC5 knockdown was barely significant, and this experiment lacks an siRNA-resistant re-expression DHHC5 control.

As explained in the manuscript, the effect of DHHC5 siRNA on phagocytosis efficiency is not necessarily due to an effect on Syk. We included these data because they are consistent with the results indicating that DHHC5 palmitoylates Syk. Nevertheless, there are many palmitoylated proteins involved in phagocytosis (Dixon *et al*, 2021), and DHHC5 can have many targets at the cup. We already answered to the siRNA-resistant DHHC5 issue.

4. Figure 3E: The inhibitory effect of massive overexpression EGFP-mDHHC20 is quite striking and a better explanation is needed for why overexpression of decreased Syk palmitoylation by 3-fold. They say "This is probably because EGFP-mDHHC20 overexpression decreased endogenous DHHC5 level by ~60 %, a drop that could be due to over-palmitoylation of mDHHC5". To be sure about this, they would need to measure the palmitoylation stoichiometry. Is DHCC5 RNA level reduced by 60% with overexpression of DHHC20? The data showing that DHHC20 palmitoylates and regulates DHHC5 levels have been published before (Plain *et al*, 2020) and a detailed study on the interconnection between DHHC5 and DHHC20 activities is beyond the scope of this paper.

Finally, the apparent decrease in Syk palmitoylation could be explained by the significantly lower level of Syk protein in the unstimulated cells compared to that in the SRBC-treated cells;

In fact, upon DHHC20 overexpression, there is less Syk in the unstimulated RAW cells but the palmitoylation level is higher. Hence, weak palmitoylation upon phagocytosis cannot be explained by Syk levels. The presented results were confirmed in another experiment and a significant decrease in Syk palmitoylation upon DHHC20 overexpression was always observed.

how could the level of Syk protein increase so rapidly upon treatment?

This difference in inputs can be due to sample handling.

5. Figure 4A: Why were so many sulfenylated bands observed in the hSyk IP sample?

Now in Fig 5A. We were indeed surprised by the number of bands. Syk likely associates with other proteins that undergoes sulfenylation. These data obtained on human primary macrophages were confirmed using RAW macrophages, and labeling was severely decreased by NAC treatment for all sulfenylated proteins (Fig. 5B). These results indicate that there might be a global regulation of the Syk signalization pathway by sulfenylation.

6. Figure 4B: The data identifying C602 as a sulfenylation site in RAW246.7 cells are not very convincing due to the fuzziness of the biotin positive bands, which have a very high smeary background, which will make it hard to accurately quantify the embedded band of interest, and therefore it is hard to be confident of the sulfenylation values that are recorded in the histogram.

Now in Fig 5C. We do not know exactly why these bands are so poorly defined. We repeated these experiments using freshly prepared reagents and obtained the same results. Please note that poorly defined bands were also observed in another sulfenylation study that used dimedone, another sulfenylation reagent (Heppner *et al*, 2018). The identification of Cys602 as an oxidized residue in mSyk is now confirmed by proteomic analyses (Fig.S6).

Moreover, the intensities of the biotinylated WT and C602S bands look similar, and, if one takes into account that the level of the C602S mutant Syk protein was significantly lower than that of the WT Syk protein, the actual degree of sulfenylation would seem to be higher for C602S mutant Syk!

Quantification showed that Syk sulfenylation is decreased by 50% upon phagocytosis for the WT while no effect was observed for the mutants.

In other words, based on these data the evidence that C602 is selectively sulfenylated does not seem to be very strong. Also, the level of the RK/AA Syk protein was extremely low especially in the unstimulated cell sample, whereas the biotinylated band was increased in this sample. Based on the quantitation below the blots, these samples were done in quadruplicate, but the authors either need to repeat this or show more convincing blots where the level of Syk protein are more consistent.

As stated before we repeated these experiments using fresh reagents and obtained the same results. It is true that bands are poorly defined, that this can be puzzling, and this is why we repeated the experiment many times. This problem is likely inherent to the detection of sulfenylated proteins (Heppner *et al.*, 2018). Statistical analysis of the results showed that significant differences were obtained. The specificity of the labeling is indicated by the NAC-mediated inhibition.

In this regard, the authors need to indicate in the legends of all the figures with samples of this sort whether these are technical or biological repeats, and how many times these experiments were repeated. We indicated that repeats correspond to entirely independent experiments.

If the current results are correct, then one also needs an explanation for why explain why the RK/AA and C591S mutants were not desulfenylated - does this process only occur when the Syk protein is recruited to the phagocytic cup, and if so what is the mechanism?

As discussed above, desulfenylation requires palmitoylation and it is thus normal that the C591S mutant is not desulfenylated. The RK/AA mutant exhibited a kind of an intermediate phenotype that is difficult to explain.

7. Figure 5: Given the relatively limited amount of information on the MDS analysis and the mechanism through which desulfenylation leads to catalytic activation, it would certainly strengthen the authors case if they had some experimental HDMX data to show that desulfenylation affects the exposure of residues 267-325 as they propose. Did the authors also carry out MDS on the S602 mutant Syk catalytic domain. What about using AlphaFold 3 for static modeling? The two 3D structure panels shown in panel B need better labeling to describe which regions of the catalytic domain we are looking at, including the Y348/Y352 residues, if they are in the model. Also, the ball and stick shapes depicting C602 and C602S-OH need to be labeled and described in the legend, and possibly an enlargement of this region shown as an inset. What specific interactions does the sulfenyl oxygen make in the model?

Please note that we performed molecular dynamics studies on human Syk (with C608, PDB 4FL3) and not mouse Syk (with C602), and that this Figure is now Fig. 6. As requested, we performed molecular dynamics studies on the C608S mutant and observed that this mutation induced a strong drop in the flexibility of the loop composed by residues 267-325. The atomic fluctuations were even lower than those observed with the sulfenylated protein. Figure 6 has been improved to allow easier identification of the different domains. The sulfenylated Cys residue is now highlighted in larger sticks or ball and sticks as requested. Concerning AlphaFold 3, it could not produce a better 3D model than that obtained with the Charmm program and, in

both cases, the flexible interdomain formed a very extended loop corresponding to residues 262 to 337, and not a folded domain. This is why we used MD for obtaining partially folded regions during the simulation, under the constraints of the force field for secondary structures.

Please note that the Tyr348/Tyr352 residues were both mutated to Phe in the construct that was used to solve the crystal structure (PDB 4FL3). We used this Syk version as a template to model the missing parts in the 4FL3 structure. The mobility of the Phe348/Phe352 residues was relatively low in all simulations as shown in Fig. 6. Regarding the interaction of the sulfenylated Cys, interestingly, we observed during molecular dynamics simulations that in WT hSyk, Cys608 is stably connected to Leu604 within the same α -helix by a hydrogen bond. Nevertheless, sulfenylation somehow destabilizes this interaction and sulfenylated Cys608 sometimes (~20% of the simulation time; Fig. S7) establishes bonding with Val555 that is localized within another α -helix. This interaction might explain the long-distance effect on the flexible interdomain. Altogether these molecular dynamics experiments indicate that sulfenylation induces a more stabilized and packed Syk structure, while desulfenylation allows molecule opening both at the level of the kinase domain and the interdomain-B.

8: Figure 6: The activity data look reasonable, but again the large variability in the levels of Syk protein within control/stimulated pairs, makes it hard to be totally convinced by the quantitation.

Now in Fig 8. Data are expressed as the Phospho-Syk/ Syk ratio and are thus independent of the overall Syk levels. Regarding the quantification, we observed a very large (75%) drop in phosphorylation for all mutants. Statistical significance is at its highest level ($p < 0.0001$).

9. Figure 7: Where is the FRET biosensor protein localized in the cells, and could this affect the interpretation of this experiment?

The probes localize essentially in the cytosol, and this is now shown in Fig. S8. Data obtained using control Syk, including the kinase-dead mutant, show that the assay is functioning.

Also, there were SRBC-stimulation induced increases in the activities of the RK/AA and C591S mutant Syk's, whereas in Figure 6 the activities of the RK/AA and C591S mutants were as low as those for the SH2 binding and C602S Syk mutants, which here showed no activation. Is there an explanation for these discrepancies?

The applied assays are different. In Fig.6 (now Fig.8) we monitored Syk phosphorylation, whereas in Fig.7 (now Fig.9) we followed the phosphorylation of a peptide that contains a Syk phosphorylation motif (from VAV2). Nevertheless, as pointed by the reviewer, it seems that the catalytic activity of the RK/AA and C591S mutant is preserved even in the absence of phosphorylation. We believe that trying to explain this difference would be largely speculative and beyond the scope of our study.

10. Figure 8: Why was the kinase-dead Syk mutant not recruited to the cup - is this because it does not activate PIP3 production? Sulfenylation and palmitoylation data for this mutant are not shown.

Now in Fig 10. We do not know exactly why the KD mutant is not recruited to the cup, but our observation indicates that the Syk catalytic activity is required for its stable association with the cup, and for the recruitment of DHHC5. For sulfenylation and palmitoylation experiments, we focused on Cys mutants.

11. Figure 10A/B: There is a huge scatter range for the WT datasets, and this means that the plots for the mutants are very compressed. Is there a better way of depicting these data?

It is now Fig.12. We checked whether data distribution was normal so that we could use ANOVA, but it is true that the distribution is wide. If we use another graphical representation, the graph will not be easily comparable with the graphs showing the recruitments of the other players (i.e. Fig.10 and 11).

12. Figure 11: Their new model is not very informative - it does not depict any conformational changes in the Syk catalytic domain upon desulfenylation or activation loop phosphorylation, and in this regard the whole Syk protein is shown as a blob without a discrete Syk catalytic domain and tandem SH2 domains, with the tandem P-ITAM phosphates that engage the tandem SH2 domains shown at either end of the Syk blob. It is now Fig.13. This Figure has been modified as requested.

Also, the associated "Src" molecule apparently has two palmitoyl groups attached, when the only c-Src lipid modification is a single N-terminal myristoyl group, with some of the other SFKs having an N-terminal myristoyl group and a palmitoylated Cys within the first 10 residues.

We fully agree, apologize for this error, and changed "Src" for "SFK".

Also, the model implies an order of Syk modification and binding. that was not fully demonstrated experimentally. Is sulfenylated Syk selectively recruited to the FcγR as implied in step 1?

Yes indeed, since the non-sulfenylable C602S is recruited to the phagocytic cup, sulfenylation plays no role in the recruitment at this level.

The model lacks the Src-mediated phosphorylation of Y348/Y352 step, whose phosphorylation initiates activation.

We agreed to improve the representation for the Syk protein (showing SH2 and kinase domains) which is the topic of this study, but space constraints hamper us to display Syk phosphorylation events that were not explored in detail in this study.

13. For other tyrosine kinases, it is known that an intramolecular S-S bond can form between Cys residues in the catalytic domain. Did the authors consider if C591 and C602 could form S-S bonds - C591 and C602 are too far apart in the Syk catalytic domain structure to form an S-S bond, but C602 might be able to form an S-S bond with another Cys, such as Cys587, which might be possible based on the crystal structure.

We will answer this point using hSyk residue numbers, based on PDB 4FL3. Disulfides can form if the distance between Cys residues is below 3.5 Å. Since the distances between Cys593 and Cys597 in the crystal structure and during the simulation are 4.3 Å and 4.6 Å, respectively, it is unlikely that S-S bonds can form between these Cys. They are also too far away from Cys608 (more than 12 Å). It should also be noted that, from a biological point of view, it is widely accepted that any disulfide bond formation in the cytosol is very transient due to the reducing environment.

Referee #2

My primary concern relates to the palmitoylation detection method. In Figure 2A, I noticed that the negative control lane (without hydroxylamine) shows comparable or even stronger signal than the experimental condition. Since the ABE assay fundamentally depends on hydroxylamine to cleave thioester bonds before biotinylation, this suggests significant non-specific binding or technical issues with the assay. This needs to be resolved, as it affects interpretation of all the palmitoylation data. Additionally, this key figure lacks quantification - it would strengthen the conclusions to show replicate experiments with statistical analysis demonstrating phagocytosis-induced palmitoylation.

The bands in the absence of hydroxylamine are very faint. We repeated this experiment many times and we can add quantification that will indeed strengthen the data. Other studies of protein sulfenylation using another reagent also results in this type of bands (Heppner *et al.*, 2018).

A similar issue with quantification appears in several other figures. For example, Figure 3F shows a co-immunoprecipitation between DHHC5 and Syk, but without quantification across multiple experiments, it's difficult to assess the significance of this interaction. Given the importance of these biochemical findings to the overall story, providing quantitative data from replicate experiments would substantially strengthen the manuscript.

We duplicated the DHHC5-Syk coimmunoprecipitation experiment (now Fig 4A) and do show the quantification. There is 11-fold more DHHC5 associated with Syk upon phagocytosis.

I'm also concerned about the biological timing of the sulfenylation measurements. The authors report Syk desulfenylation at 5 minutes post-phagocytosis, but this precedes the major oxidative burst, which typically peaks around 30-60 minutes when NADPH oxidase is fully active.

This study focuses on the initial event of phagocytosis (up to phagosome closure, ~5 min) but it was effectively interesting to examine whether Syk sulfenylation / desulfenylation could be correlated to H₂O₂ production at the cup at the onset of phagocytosis. We took advantage of a sensitive and specific H₂O₂ reporter (plifeAct-Hyper7 (Pak *et al*, 2020)) that allowed us to observe an oxidative burst at the onset of phagocytosis (after 160-180sec, new Figure 7 and video 1). H₂O₂ production severely drops after 200 sec. Since H₂O₂ needs 2 min to activate Syk (Qin *et al.*, 1998), these H₂O₂ production results are consistent with the observation that Syk is desulfenylated 5 min after the onset of phagocytosis.

This raises important questions: Is the observed desulfenylation related to loss of basal modification rather than oxidative burst-dependent changes? It would be valuable to extend the time course to capture the actual oxidative burst period and include NADPH oxidase inhibitors to test ROS dependence. This would help clarify the biological context of the modification.

Many enzymes are able to produce ROS (Altenhofer *et al*, 2015). We believe that our new data presented in the manuscript (the H₂O₂ wave-like production, Fig.7 and videos, and the inhibitory effect of NAC on Syk sulfenylation, Fig.5B) are sufficient to complete this study.

The manuscript would also benefit from more precise language regarding the necessity of these modifications. While the authors state that palmitoylation is "strictly necessary" for phagocytosis, Figure 1B shows that the C591S mutant retains approximately 30% activity. This suggests the modification is important for optimal function rather than absolutely required. Adjusting the language throughout to reflect these partial phenotypes would provide a more accurate representation of the data.

There are only two occurrences of "strictly" in the manuscript. The first one is in the introduction, Syk is strictly need for phagocytosis, and the second concerns the C602S mutant that does not show any significant phagocytic activity (Fig.1B). We tried to interpret our results as objectively as possible.

An important technical consideration is the reduced expression of the C591S and C602S mutants shown in Figure S2. Since these mutants express at substantially lower levels than wild-type, this could contribute to their functional deficits. It would be helpful to either normalize functional data to expression levels or discuss how stability effects might be part of the regulatory mechanism for these modifications.

In biochemical experiments, we always normalize data using input data (i.e. Syk levels). For imaging we always select cells that display equivalent levels of Syk-GFP (see Fig. 10 for instance). We agree that this is an important point. This is now specified in the Methods section.

The siRNA experiments in Figure 3 would be strengthened by including rescue controls. While the knockdown data suggests DHHC5 involvement, the partial knockdown and compensatory DHHC20

upregulation complicate interpretation. Rescue with an siRNA-resistant DHHC5 would provide stronger evidence for specificity.

As detailed in the response to Reviewer 1, we do not believe that it is necessary to add additional controls, as we used a combination of 4 siRNAs. Moreover, ectopic expression of DHHC20 decreases DHHC5 level and Syk palmitoylation just as siRNAs targeting DHHC5 (Fig.3), thereby confirming the DHHC5 siRNA effect. To our opinion, ample evidence has already been provided that Syk palmitoylation is performed by DHHC5, *i.e.* siRNA-mediated extinction, overexpression, co-IP and accumulation at the phagocytic cup.

The unexpected finding that DHHC20 overexpression decreases palmitoylation (Figure 3E) is interesting and might warrant further investigation into the regulatory relationships between these enzymes.

As explained above, and indicated in the discussion section, the relationship between these enzymes has already been published (Plain *et al.*, 2020).

The molecular dynamics simulations, while providing some structural insights, are limited by the absence of membrane context and the relatively short timescale. Since palmitoylation mediates membrane association, including membrane in the simulations would provide more physiologically relevant information.

Alternatively, the limitations of the current simulations should be acknowledged more explicitly.

Regarding the interaction of Syk with the membrane, the palmitoylation that takes place on the hSyk-Cys597 residue allows the anchoring of Syk to the membrane. However, this palmitoylated cysteine is localized at the opposite side of the Syk interdomain B (a highly flexible region in WT Syk) that is thus unlikely to interact with the membrane. Regarding the timescale, the 500 ns length we used is thought to be sufficient to predict remodeling of the protein (Yu *et al.*, 2022), and this was our main goal. It was nevertheless too short to observe any structuration of the interdomain B. This is now indicated in the paper.

Additionally, there are inconsistencies between assays (how can C591S have 30% phagocytosis but almost no F-actin accumulation?),

These are experimental results from different type of assays, time and phagocytic targets. F-actin accumulation is monitored after 5 min on opsonized SRBCs, while phagocytosis is observed after 15 min (to allow multiple phagocytosis events) using opsonized beads. The C591S mutant might just slow down actin polymerization.

the poor quality of the sulfenylation blots,

As indicated before, we repeated these experiments several times using freshly prepared reagents and obtained the same results. This problem is likely inherent to the detection of sulfenylated proteins (Heppner *et al.*, 2018). Statistical analyses of our data supported their significance. The identification of Cys602 as oxidized residue in mSyk is now confirmed by proteomic analyses (Fig. S6).

and the oversimplified model in Figure 11 that ignores all the partial phenotypes.

Now in Fig 13. This is a model that obviously summarizes and simplifies the message.

Referee #3

General comment: the figures could be re-organised to be more compact and leave less white spaces in order to give a better general overview of the results.

In fact the figures were made for initial manuscript reviewing. We now provide more compact high-resolution figures.

Specific comments on the figures

Figure 1A: the authors use SYK knock-out cells: how many clones were used in the study?

As now stated in the Methods section, we generated 7 clones and selected one to be used in the experiments. Fig.1 shows that Syk is specifically deleted and that the phagocytic function can be restored by ectopic Syk expression.

Figure 1B: there seem to be only 2 data points in the graph: how were the statistics generated? Figure 3E: same question

These are two entirely independent experiments that provided identical results. We used ANOVA for statistics.

Figure 3C: the quantification performed is not well described and the legend of the Y axis is also unclear: what does +/- phago mean?

We explain this point in more details in the methods section. We first divide the ABE values by the input values, then calculate the ratio (with phago) / (without phago).

Figure 8: the recruitment of SYK at the phagocytic cup in the WT and RK/AA conditions does not seem to be a very local event, as one can tell from the image presented. Could the authors comment on this? In addition, image analysis and quantification of the recruitment was performed on 6-9 cells only, and it is not mentioned from how many independent experiments. This result should be improved with more data points. Now in Fig 10. We obtained statistically significant differences among mutants, indicating that the number of cells we used is sufficient.

Figure 8, 9, 10: could the authors provide the blue images in cyan or better, in black and white to improve their quality?

These figures (now 10,11,12) have been modified as requested.

Figure 11: there is a problem of scale for the Ig and the red blood cell. As some experiments were performed with beads, the summary scheme could be represented with only a portion of a larger bead/ particle. More importantly, the Fc portion does not bind to the FcR as it is represented. The representation of the results obtained in the manuscript should be indicated with colors different from the mechanisms that are still hypothetical.

We corrected this Figure (now Fig 13) as requested by the reviewer.

Minor comments:

- RAW cells should be RAW264.7 cells in the text

This was corrected, although this abbreviation is widely used in papers.

There are some typos :

- p4 Line 29: we observed using that DHHC5 specifically associates with Syk

- p4 the title: DDHC5: should be replaced by DHHC5

- supp fig 4 to 7 legends: Representative line plots (width: ten pixels) across cells, enabling to monitor XXX compared to a non-phagocytic region

Thank you, these points were rectified.

I look forward to hearing from you on our revised manuscript.

Best regards

Bruno Beaumelle, PhD

Team leader IRIM

References

- Altenhofer S, Radermacher KA, Kleikers PWM, Wingler K, Schmidt HHHW (2015) Evolution of NADPH Oxidase Inhibitors: Selectivity and Mechanisms for Target Engagement. *Antioxid Redox Signal* 23: 406-427
- Chopard C, Tong PBV, Toth P, Schatz M, Yezid H, Debaisieux S, Mettling C, Gross A, Pugnieri M, Tu A *et al* (2018) Cyclophilin A enables specific HIV-1 Tat palmitoylation and accumulation in uninfected cells. *Nature communications* 9: 2251
- Dixon CL, Mekhail K, Fairn GD (2021) Examining the Underappreciated Role of S-Acylated Proteins as Critical Regulators of Phagocytosis and Phagosome Maturation in Macrophages. *Front Immunol* 12: 659533
- Gao X, Hannoush RN (2018) A Decade of Click Chemistry in Protein Palmitoylation: Impact on Discovery and New Biology. *Cell Chem Biol* 25: 236-246
- Hepner DE, Dustin CM, Liao C, Hristova M, Veith C, Little AC, Ahlers BA, White SL, Deng B, Lam Y-W *et al* (2018) Direct cysteine sulfenylation drives activation of the Src kinase. *Nature communications* 9: 4522
- Krasnowska EK, Pittaluga E, Brunati AM, Brunelli R, Costa G, De Spirito M, Serafino A, Ursini F, Parasassi T (2008) N-acetyl-l-cysteine fosters inactivation and transfer to endolysosomes of c-Src. *Free Radic Biol Med* 45: 1566-1572
- Pak VV, Ezerina D, Lyublinskaya OG, Pedre B, Tyurin-Kuzmin PA, Mishina NM, Thauvin M, Young D, Wahni K, Martinez Gache SA *et al* (2020) Ultrasensitive Genetically Encoded Indicator for Hydrogen Peroxide Identifies Roles for the Oxidant in Cell Migration and Mitochondrial Function. *Cell Metab* 31: 642-653.e646
- Paulsen CE, Truong TH, Garcia FJ, Homann A, Gupta V, Leonard SE, Carroll KS (2011) Peroxide-dependent sulfenylation of the EGFR catalytic site enhances kinase activity. *Nat Chem Biol* 8: 57-64
- Plain F, Howie J, Kennedy J, Brown E, Shattock MJ, Fraser NJ, Fuller W (2020) Control of protein palmitoylation by regulating substrate recruitment to a zDHHC-protein acyltransferase. *Commun Biol* 3: 411
- Qin S, Kurosaki T, Yamamura H (1998) Differential regulation of oxidative and osmotic stress induced Syk activation by both autophosphorylation and SH2 domains. *Biochemistry* 37: 5481-5486
- Werno MW, Chamberlain LH (2015) S-acylation of the Insulin-Responsive Aminopeptidase (IRAP): Quantitative analysis and Identification of Modified Cysteines. *Sci Rep* 5: 12413
- Yu Y, Wang Z, Wang L, Tian S, Hou T, Sun H (2022) Predicting the mutation effects of protein-ligand interactions via end-point binding free energy calculations: strategies and analyses. *J Cheminform* 14: 56
- Yucel BP, Henley JM, Wilkinson KA (2024) Protocol for detecting palmitoylation of high-molecular-weight rat synaptic proteins via acyl-PEG labeling. *STAR Protoc* 5: 103296

December 24, 2025

RE: Life Science Alliance Manuscript #LSA-2025-03500-TR

Dr. Bruno Beaumelle
Institut de Recherche en Infectiologie de Montpellier
Université de Montpellier-CNRS
1919 route de Mende
MONTPELLIER 34293
France

Dear Dr. Beaumelle,

Thank you for submitting your revised manuscript entitled "Syk activation during FcγR-mediated phagocytosis involves Syk palmitoylation and desulfenylation". We returned the manuscript to the original Reviewers 1 and 3, whose reports are below.

As you will see, both reviewers remark on the several changes and additional observations included in this revision. We agree with Reviewer 1 that the normalization shown in Figure 2B should be clearly explained in the figure legend. In accordance with LSA policies on figure preparation, divisions between separate blots must be clearly indicated, as requested also by Reviewer 1. We agree that the source of cells used for mass spec analysis should be briefly stated in the results section related to Fig S6, as described already in the methods. Finally, Reviewer 3 suggested changes to the graphical abstract, which we leave to your discretion. We would be happy to publish your paper in Life Science Alliance pending these changes and final revisions necessary to meet our formatting guidelines.

- Please be sure that the authorship listing and order is correct.
- Please upload your Table in editable .doc or Excel format.
- Please add the X and Bluesky handles of your host institute/organization, as well as your own and/or one of the authors, in our system.
- Please add a Conflict of Interest statement to your main manuscript text.
- Please add an Author Contributions section to your main manuscript text.
- A "Data Availability" section should be placed after the Materials & Methods section. Please consult our guidelines at <https://www.life-science-alliance.org/manuscript-prep#format>
- Table should be numbered with Arabic numerals (1, 2, 3, 4).
- Please remove legends from the supplementary figures. Legends should appear only in the manuscript file.
- Please add your main, supplementary figure, table, and video legends to the main manuscript text after the references section.
- Please add callouts for Figures 6D, E and S3A, B to your main manuscript text.
- Please add molecular weight markers to the gel shown in Fig 1.
- We encourage you to remove Fig 13 and upload this image as a Graphical Abstract file. Please be aware that this would remove it from the manuscript pdf.

A. FINAL FILES:

B. MANUSCRIPT ORGANIZATION AND FORMATTING:

Thank you for your attention to these final processing requirements. Please revise and format the manuscript and upload materials as soon as you are able.

Sincerely,

Reviewer #1 (Comments to the Authors (Required)):

To address the reviews of the original paper, the authors have done some new experiments and added some new figures to address the reviewers' points. In particular, they have used MS analysis to provide direct evidence that Cys602 is oxidized to the Cys602-SO₂H sulfinic acid state, possibly as a result of spontaneous oxidative of C602-SOH sulfenic acid to C602-SO₂H during sample preparation (new Figure S6). They have also partially addressed issues with the previous variability in the Syk protein signal by including some new blots.

1. New Figure 2B: It is not clear whether the palmitoylation signal was normalized to the level of Syk protein - if not, it needs to be. In addition, It does not look as though these new samples were run on the same gel - separations are shown between the WT and the RK/AA pairs of lanes, as well as between these lanes and the four C591S and C602S lanes on the right. This needs to be explained in the figure legend. Also, the title of the Figure 2 legend needs to be corrected - currently it is: Figure 2. Syk is palmitoylated on mSyk-Cys501.

2. New Figure 5B: The new data showing that NAC pretreatment reduced sulfenylation are not very persuasive, particularly, because there are so many bands in these lanes. Also, in this panel the control sample has a biotin-Syk/Syk ratio greater than 1 - is this because the blots were exposed for different times.

3. New Figure 6: It would help the reader if panels C, D and E were labeled on the figure itself with the protein that was modeled,

possibly highlighting regions that differ, including the 262 to 337 loop, and C608.

4. New Figure S5: Neither the legend nor the text indicated what cells or conditions were used to generate the Syk sample used for MS analysis!

7. New Figure 13: This new model is improved, now showing that Syk palmitoylation occurs after ITAM engagement with the phospho-FcγR prior to desulfenylation. What is still unclear is what happens upon desulfenylation that promotes PIP3 binding and SFK-mediated Y525/526 phosphorylation.

Reviewer #3 (Comments to the Authors (Required)):

The manuscript by Jansen et al., describes novel mechanisms of activation of the tyrosine kinase SYK that plays an important role downstream of the FcγR-mediated phagocytosis. In particular the study focuses on two post-translational modifications, palmitoylation and sulfenylation. Palmitoylation of SYK is shown to be mediated by the S-acyl transferase DHHC5 and to be necessary for SYK recruitment to the phagocytic cup, activation and downstream events such as Cdc42 activation and F-actin polymerisation. Sulfenylation was proposed to play an inhibitory role as desulfenylation occurs upon FcR-phagocytosis and molecular dynamics experiments suggested that it could facilitate phosphorylation of the kinase.

This is a very interesting manuscript that really extends our knowledge on the modalities of regulation of a kinase that is crucial in many phagocytic processes.

The authors have improved the manuscript according to my comments and the other reviewers' comments on a previous version of the paper.

However, they have not modified the graphical abstract: there is a problem of scale for the Ig and the red blood cell. As some experiments were performed with beads, the summary scheme could be represented with only a portion of a larger bead/particle. More importantly, the Fc portion does not bind to the FcR as it is represented. The representation of the results obtained in the manuscript should be indicated with colors different from the mechanisms that are still hypothetical.

January 13, 2026

RE: Life Science Alliance Manuscript #LSA-2025-03500-TRR

Dr. Bruno Beaumelle
Institut de Recherche en Infectiologie de Montpellier
IRIM-CNRS
1919 route de Mende
MONTPELLIER 34293
France

Dear Dr. Beaumelle,

Thank you for submitting your Research Article entitled "Syk activation during FcγR-mediated phagocytosis involves Syk palmitoylation and desulfenylation". It is a pleasure to let you know that your manuscript is now accepted for publication in Life Science Alliance. Congratulations on this interesting work.

DISTRIBUTION OF MATERIALS:

Again, congratulations on a very nice paper. I hope you found the revision process to be constructive and are pleased with how the manuscript was handled editorially. We look forward to future exciting submissions from your lab.

Sincerely,
